# BRD2 regulation of sigma-2 receptor upon cholesterol deprivation

Hongtao Shen[1],*, Jing Li[1],*, Xiujie Xie[1], Huan Yang[1] , Mengxue Zhang[1], Bowen Wang[1], K Craig Kent[1], Jorge Plutzky[2], Lian-Wang Guo[1,3]

The sigma-2 receptor (S2R) has long been pharmacologically targeted for antipsychotic treatment and tumor imaging. Only recently was it known for its coding gene and for its role implicated in cholesterol homeostasis. Here, we have investigated the transcriptional control of S2R by the Bromo/ExtraTerminal epigenetic reader family (BETs, including BRD2, 3, and 4) upon cholesterol perturbation. Cholesterol deprivation was induced in ARPE19 cells using a blocker of lysosomal cholesterol export. This condition up-regulated S2R mRNA and protein, and also SREBP2 but not SREBP1, both transcription factors key to cholesterol/fatty acid metabolism. Silencing BRD2 but not BRD3 or BRD4 (though widely deemed a master regulator) averted S2R up-regulation that was induced by cholesterol deprivation. Silencing SREBP2 but not SREBP1 diminished S2R expression. Furthermore, endogenous BRD2 co-immunoprecipitated with the transcription-active N-terminal half of SREBP2, and chromatin immunoprecipitation-qPCR signified co-occupancy of BRD2, H3K27ac (histone acetylation), and SREBP2Nterm at the S2R gene promoter. In summary, this study reveals a previously unrecognized BRD2/SREBP2 cooperative regulation of S2R transcription, thus shedding new light on signaling in response to cholesterol deprivation.

## Introduction

After decades of studies, cholesterol biology remains inadequately understood, in particular, the regulations involving lysosomes, which distribute cholesterol to other organelles (1). Recently, TMEM97 was reported as a novel player in cholesterol transport (2, 3). Though an ER resident protein, TMEM97 can translocate to the lysosomal membrane where it appears to attenuate the activity of NPC1 (Niemann–Pick disease, type C) (3), the transporter that "pumps" cholesterol out of the lysosome. TMEM97 was also suggested to interact with LDLR thereby involved in cholesterol uptake (3). Intriguingly, the coding gene of the sigma-2 receptor (S2R), an enigmatic drug-binding site pharmacologically identified 40 yr ago, was finally (in 2017) unveiled to be TMEM97 (4).

There are a very small number of articles published on TMEM97 (also known as MAC30) (5); studies on S2R are largely limited to pharmacology such as anti-psychotic treatments (6). As a result, little is known about the molecular regulations of S2R/TMEM97/MAC30 (hereafter denoted as S2R for clarity) (4). S2R is highly expressed in progressive tumors and thereby targeted as a biomarker for diagnostic imaging (7). Silencing S2R appeared to alleviate Niemann–Pick disease condition in a mouse model, which features mutated NPC1 and consequential cholesterol accumulation in lysosomes (2). It is, thus, important to understand how S2R expression is controlled, whereas little is known at present.

Gene transcription programs are coordinately governed by transcription factors (TFs) and epigenetic factors. Rapidly growing literature supports a notion that the family of BETs (Bromo/ExtraTerminal-domain proteins) act as epigenetic determinants of transcription programs (8). Among the BETs (BRD2, 3, and 4), BRD4 is best studied and thought to function as a lynchpin-organizer of transcription assemblies. Whereas its C-terminal domain (absent in BRD2 and BRD3) interacts with the transcription elongation factor (pTEFb) that activates the RNA polymerase II, its two bromodomains read (bind) acetylation bookmarks on histones and TFs. These interactions usher a transcription assembly to specific genomic loci (9). BRD4 has been shown to play a master role in broad cellular processes ranging from proliferation to differentiation (10, 11) to autophagy (12). Curiously, whether BETs are directly involved in cholesterol homeostasis remained unclear (13). Relevant to this question, recent studies identified a crucial role of BRD4 in adipogenesis (10). We were thus encouraged to explore whether BRD4 regulates the expression of S2R, a novel modulator of cholesterol transport (2).

Herein, we found that pan-BETs inhibition abolished S2R up-regulation that was induced by cholesterol deprivation. However, it

[1]Department of Surgery, School of Medicine, University of Virginia, Charlottesville, VA, USA    [2]Cardiovascular Division, Brigham and Women's Hospital, Harvard Medical School, Boston, MA, USA    [3]Robert M. Berne Cardiovascular Research Center, University of Virginia, Charlottesville, VA, USA

Correspondence: lg8zr@virginia.edu
*Hongtao Shen and Jing Li contributed equally to this work

was BRD2 but not BRD4 or BRD3 that was responsible for this BET function. This was unexpected given that BRD4 has been widely reported to be the determinant BET in diverse processes (8). We also found that BETs inhibition repressed the transcription of both SREBP1 and SREBP2, the master TFs governing fatty acid and cholesterol homeostasis, and silencing SREBP2 but not SREBP1 inhibited S2R expression. Further results suggested a novel mechanism whereby BRD2/SREBP2 co-occupy promoter regions of the S2R gene activating its transcription. Considering that BETs and S2R are targets of clinical (or trial) drugs (6, 14), in particular, with their newly discovered involvement in SARS-CoV-2 infection (COVID-19) (15), our findings may impact research not only in cholesterol biology but also translational medicine.

## Results

### Pan-BETs inhibition represses S2R expression that is stimulated by cholesterol deprivation

To study regulations of S2R, we chose ARPE19, a human epithelial cell line where cholesterol plays a critical role in cellular function/dysfunction (16). To establish a cellular model to monitor S2R level changes, we first tested commonly used cytokine stimulants, including PDGF-AA, PDGF-BB, and TGFβ1. However, they did not significantly alter S2R mRNA levels (Fig S1). We then used U18666A (abbreviated as U18 throughout) as a tool to generate an experimental condition for cholesterol deprivation, as it is an established NPC1 inhibitor that keeps cholesterol trapped inside the lysosome thereby producing an intracellular environment with cholesterol reduced in the ER (17 Preprint, 18). We found that treatment with U18 increased S2R mRNA by up to eightfold and S2R protein by ~ 2-fold. NPC1 silencing did not produce this effect (Fig S2). More interestingly, pre-treatment with JQ1 (first-in-class BETs-selective inhibitor) (19, 20) abrogated this U18-induced S2R up-regulation (Fig 1A and B) (for the full blots, please see Source Data). The JQ1's enantiomer, which is a chemically identical yet functionally inert stereoisomer hence an ideal negative control for JQ1 (11, 20), did not significantly alter S2R expression (Fig S3), confirming the JQ1 effect's specificity for BETs. Serum starvation had a relatively (versus U18) minor effect on S2R mRNA and protein up-regulation, as seen in Fig 1C and D. We also performed ex vivo starvation treatment of mouse eyes, and the qRT-PCR result from primary retinal pigment epithelial cells was essentially the same (Fig 1E). Confirming the function of U18 in reducing cholesterol in the ER (3, 18), filipin staining of cholesterol disappeared from most of the intracellular space (or the ER network) after U18 treatment but instead accumulated in perinuclear structures (Fig 1I and J), which are typically known to be lysosomes (3).

Taken together, whereas ER cholesterol deprivation dramatically elevated S2R expression, pan-BETs inhibition with JQ1 abolished this up-regulation.

### Silencing BRD2 but not BRD4 or BRD3 reduces S2R mRNA and protein

JQ1 is highly selective to the BET family yet it is a pan inhibitor that blocks the bromodomains in all BETs (20). We therefore next

determined by gene silencing which BET mainly accounted for the U18-induced S2R up-regulation. Given a wealth of literature evidence indicating BRD4 as a powerful regulator in a broad range of processes (21), we expected BRD4 to be the determinant BET. To our surprise, silencing neither BRD4 nor BRD3 affected U18-induced S2R mRNA expression; rather, silencing BRD2 markedly reduced S2R mRNA levels (Fig 1F–H). Further supporting this result, silencing of BRD2 but not BRD4 or BRD3 reduced U18-elevated S2R protein levels (Fig 2A–C).

It is noteworthy that BRD4 silencing appeared to increase BRD2 protein and vice versa though without reaching statistical significance (Fig 2A and C). An interesting question thus arose as to whether BRD4 silencing could have raised S2R levels by increasing BRD2 (Fig 2C). We then silenced BRD2 in addition to BRD4 (Fig 2D), and indeed, the S2R level change became no longer evident. The simplest explanation is that the changes of S2R toward opposite directions caused by silencing BRD2 and silencing BRD4 "canceled" each other. Therefore, this result from BRD2 and BRD4 double silencing further supports a positive role for BRD2 in regulating S2R expression.

### BRD2 but not BRD4 or BRD3 gain-of-function increases S2R mRNA and protein

To further delineate a BRD2-specific role in governing S2R expression, we performed gain-of-function experiments using vectors to overexpress BRD2, BRD3, or BRD4; each vector prominently increased its respective BET member expression at both mRNA and protein levels (Figs 3 and 4). Whereas increasing BRD2 further enhanced U18-induced up-regulation of S2R mRNA and protein (Figs 3A and 4A), BRD3 or BRD4 gain-of-function did not show this effect (Figs 3 and 4). It is interesting to note that BRD4 overexpression significantly reduced BRD2 protein, and S2R protein as well (Fig 4C), consistent with a positive role for BRD2 in the regulation of S2R expression. Furthermore, protein levels of both BRD2 and BRD4 significantly decreased because of BRD3 overexpression; yet, the S2R protein level did not change (Fig 4B). This result echoed that of BRD2 and BRD4 double silencing (Fig 2D). Thus, while revealing the influence of BET proteins for each other, the gain-of-function experiments further confirmed a specific role for BRD2 in positively regulating S2R expression levels.

### Silencing SREBP2 but not SREBP1 represses S2R mRNA and protein expression

It is established that SREBP2 in complex with the SREBP-cleaving activation protein can sense a decrease of ER cholesterol and translocate to Golgi, where the SREBP2 N-terminal half (abbreviated as SREBP2Nterm) is cleaved off and then able to enter the nucleus to bind sterol regulatory elements (SREs) of genomic DNA, thereby regulating gene expression (22). In a previously reported RNAi screening study, S2R (TMEM97) was found to be a target gene of the TF SREBP2 (3). However, detailed studies on transcriptional regulation of S2R were not available, and whether S2R is also under the control of the functionally paired TF SREBP1 was not known. We thus performed silencing of SREBP2 and SREBP1 to determine their function in S2R expression. As shown in Fig 5A, whereas S2R mRNA

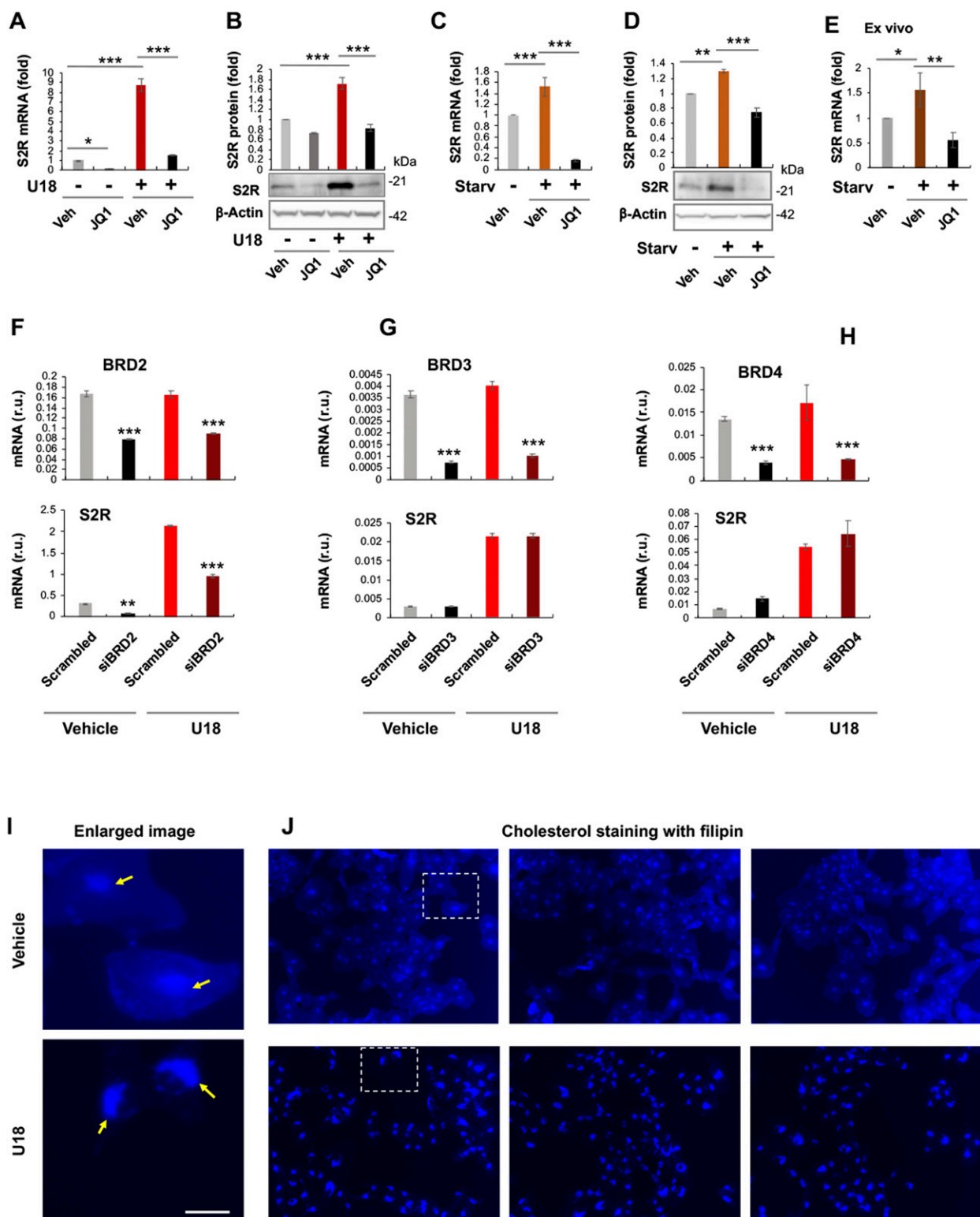

**Figure 1.   Pan-BET inhibition prevents sigma-2 receptor (S2R) up-regulation upon cholesterol deprivation.**
**(A, B, C, D)** Effect of JQ1 pretreatment of ARPE19 cells on S2R mRNA and protein levels. **(E)** Effect of ex vivo JQ1 treatment of mouse eyecups. **(F, G, H)** BRD2 but not BRD3 or BRD4 silencing reduces S2R mRNA levels in ARPE19 cells. **(I, J)** Filipin staining of cholesterol. Enlarged boxes from (J) are shown in (I); arrows point to perinuclear lysosomes; the nuclei are profiled by ring-like structures. Note that without U18 treatment, filipin staining appeared as blue haze in the cytosol (or ER network), which diminished after U18 treatment and instead accumulated in the perinuclear lysosomes. Scale bar: 5 $\mu$m. ARPE19 cells were cultured to an ~70–80% confluency in the DMEM/F12 medium containing 10% FBS. The cells were transfected with a scrambled or specific siRNA, and then incubated with U18666A (abbreviated as U18, final 5 $\mu$M)

levels were elevated by either U18 or starvation (to a minor degree), SREBP2 silencing diminished this up-regulation. Similar results occurred at the protein level (Fig 5B); that is, SREBP2 silencing abrogated U18-induced S2R protein up-regulation, confirming TF SREBP2 as a positive regulator of S2R. Indeed, ChIP-qPCR assay consistently showed SREBP2Nterm occupancy at the S2R gene promoter (Fig S4). In contrast to SREBP2, SREBP1 silencing did not reduce but rather, further enhanced S2R mRNA expression (Fig 5A). The mRNA result was largely reproduced at the protein level albeit the increase of S2R protein due to SREBP1 silencing did not reach a statistical significance (Fig 5C).

Now that the above results showed that both BRD2 and SREBP2 were positive regulators of S2R expression, we next used combined siRNAs for BRD2 and SREBP2 double silencing. The data (Fig 6) demonstrated that BRD2/SREBP2 double silencing nearly completely blocked S2R protein production. Therefore, up to this point, our results had provided compelling evidence for BRD2 being a positive regulator of S2R mRNA and protein expression.

### Pan-BETs inhibition reduces SREBP2 and SREBP1 expression levels

Given that S2R expression was controlled by both BRD2 and SREBP2, an epigenetic factor and a TF, respectively (Figs 1–6), we next asked whether BRD2 regulated S2R expression via SREBP2. As shown in Fig 7A–D, treatment with U18 increased SREBP2 mRNA (by ~4-fold) and protein (albeit statistically insignificant), and SREBP1 mRNA to a minor extent. In contrast to U18, the starvation condition up-regulated both SREBP2 and SREBP1 mRNA (by 10 to 15-fold) and protein (Fig 7E–H). Remarkably, pretreatment with JQ1 abolished all of these changes. Moreover, using a different cell line (HEK293), we also observed up-regulation of SREBP2 and S2R in U18-treated cells and its abolishment by pretreatment with JQ1 (Fig 7I–K). Consistently, SREBP2 promoter DNA co-IPed with BRD2 in ARPE19 cells (Fig S5). These results suggest that the BETs family may play a role in the transcription of SREBP2 and SREBP1, two key TFs in cholesterol and fatty acid metabolism. To the best of our knowledge, regulations of SREBPs by BETs have not been previously determined in the specific context of cholesterol perturbation.

We then determined whether SREBPs reciprocally influence BRD2 levels. The data indicated that SREBP2 silencing did not make a significant difference in either BRD2 mRNA or protein expression (Figs 8A and S6). However, SREBP1 silencing appeared to increase BRD2 protein (albeit with no statistical significance, Fig 8B), consistent with the above observation that SREBP1 silencing led to increased S2R expression (Fig 5). Taken together, silencing SREBP2

reduced S2R expression (Figs 5 and 6) without altering BRD2 levels (Fig 8A), suggesting that SREBP2 is not upstream but possibly downstream of BRD2. Interestingly, Fig 8C and D indicated that neither BRD2 nor BRD4 silencing significantly altered SREBP2 or SREBP1 protein levels. This lack of changes in SREBP protein levels may be rationalized by post-transcriptional regulations.

### BRD2 co-immunoprecipitates with the SREBP2 transcription-active N-terminal domain

Accumulating evidence suggests that BETs cooperate with specific TFs to assume transcriptional activation of select sets of genes, by two possible ways: altering the TF protein level or forming a complex with the TF to activate the transcription of target genes. Since the former did not occur (Fig 8), we next determined whether BRD2 forms a complex with the SREBP2 protein in the regulation of S2R transcription (Fig 9) (for the full blots, please see Source Data). Indeed, our data indicated that endogenous BRD2 co-immunoprecipitated with the SREBP2 N-terminal half molecule (Fig 9A), which is known to be the active form of SREBP2 that is able to translocate into the nucleus to assume the SREBP2 TF function (3). The specificity of this SREBP2Nterm/BRD2 co-IP is evident in the following observations. (1) BRD2 did not co-IP with FLAG, the empty-vector control (Fig 9A); (2) BRD2 (Fig 9A and B) but not BRD4 (Fig S7) co-IPed with SREBP2Nterm; (3) the BRD2/SREBP2Nterm co-IP was further enhanced by U18 treatment, although the change did not reach a statistical significance (Fig 9B), reflecting a likelihood that BRD2 was nearly saturated by the binding of overexpressed SREBP2Nterm. These results together suggest that the transcription active form of SREBP2, namely, SREBP2Nterm, forms a protein complex with BRD2.

### BRD2 immunoprecipitates SREBP-binding DNA regions of the S2R gene promoter

Finally, we used a BRD2 antibody for IP and performed ChIP-qPCR to detect S2R promoter regions that contain predicted SREs or consensus SREBP-binding motifs (3). We found that whereas U18 treatment increased qPCR signal of an S2R promoter region (~1,000 bp from TSS) by > 2-fold, increasing BRD2 further augmented the signal, either in the absence or presence of U18 (Fig 9C). Similar results were observed in the experiments detecting other two co-IP'ed DNA segments containing predicted SREs (Fig S8). The specificity of the ChIP-qPCR assay was confirmed by an outstanding signal-to-background ratio; that is, a large (~6 to 10-fold) difference in qPCR reading between the experiments using a BRD2 antibody and IgG control for ChIP (see Fig S8). Moreover, we performed the

for 24 h before qRT-PCR or Western blot assay. For starvation treatment, the medium was changed, which contained 0% FBS instead. For inhibition of BETs, JQ1 (1 $\mu$M) or vehicle control (equal amount of DMSO) was included in the cell culture during the treatment with U18 or starvation. For the ex vivo experiment (E), mouse eyes were dissected to remove the retina and expose the retinal pigment epithelium layer. Eyecups were incubated in the HBSS buffer with or without 10% FBS (i.e., starvation) for 24 h, and retinal pigment epithelium cell total RNA was extracted from the eyecups using the Trizol reagent for qRT-PCR assay. Quantification: At least three independent repeat experiments were performed; data were normalized to GAPDH (qRT-PCR) or $\beta$-actin (Western blot) and then to the basal control (vehicle, no U18, no starvation). The normalized data were averaged (n ≥ 3) to calculate mean ± SEM. For the ex vivo experiments, data from three mice were averaged (mean ± SEM, n = 3). Each plot in (F, G, H) represents one of two similar experiments (mean ± SD, n = 3 replicates in the same experiment). Statistics: one-way ANOVA with Bonferroni post hoc test; *$P < 0.05$, **$P < 0.01$, ***$P < 0.001$. For simplicity, non-significant difference is not labeled throughout the figures; r.u., relative units.
Source data are available for this figure.

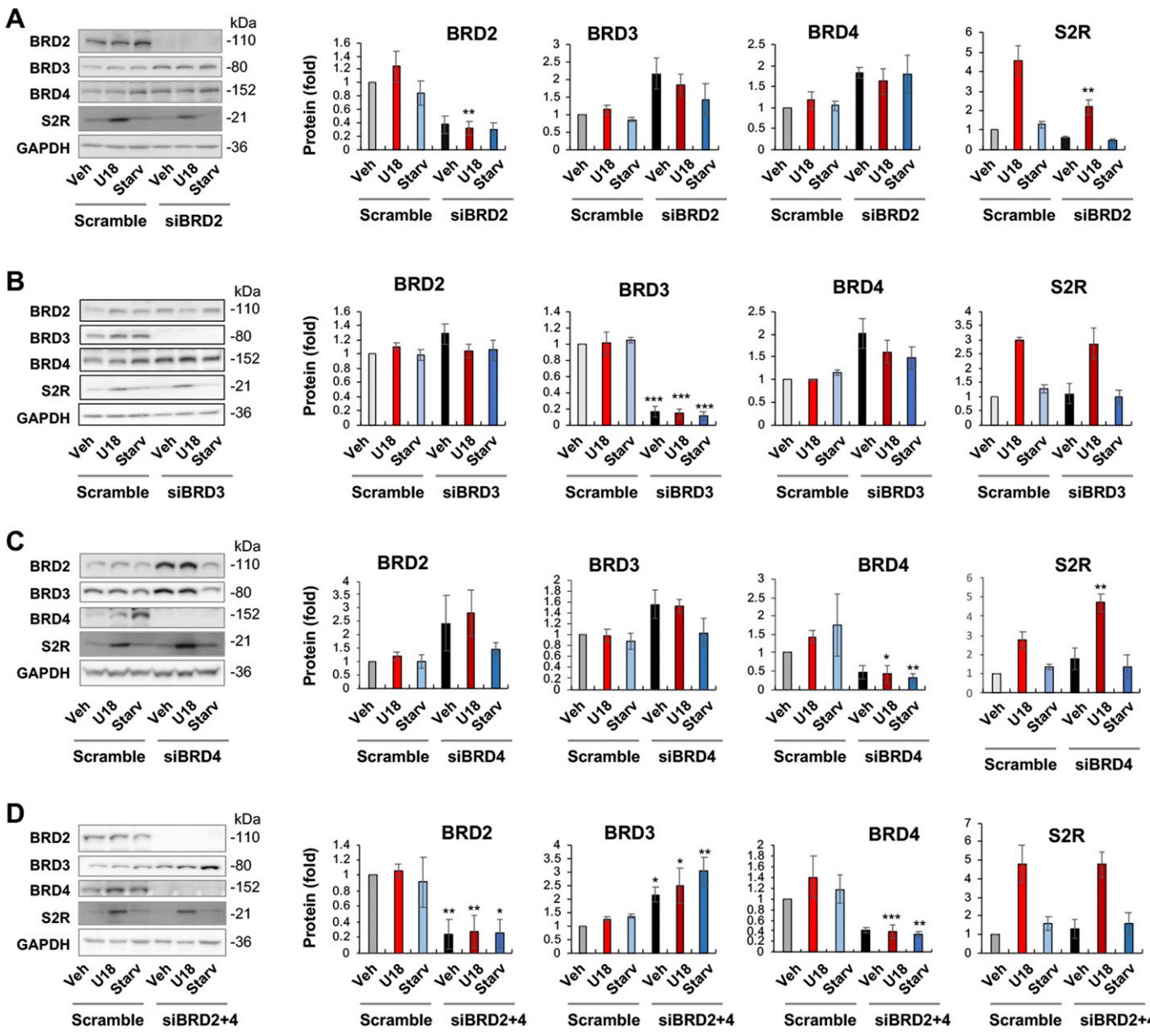

**Figure 2. BRD2 but not BRD3 or BRD4 silencing reduces sigma-2 receptor protein.**
**(A, B, C)** BRD2, BRD3, and BRD4 silencing, respectively. **(D)** BRD2 and BRD4 double silencing. ARPE19 cells were transfected with a scrambled (control) or specific siRNA for BRD2, BRD3, or BRD4 in DMEM/F12 containing 10% FBS. The cells were then treated with U18 (or vehicle) or starvation (DMEM containing 0% FBS) for 24 h before harvest for Western blot analysis. Quantification: at least three independent repeat experiments were performed; densitometry was normalized to GAPDH and then to the basal control (vehicle, scrambled). The normalized data were averaged to calculate mean ± SEM. Statistics: one-way ANOVA with Bonferroni post hoc test; n ≥ 3 independent repeat experiments; *$P < 0.05$, **$P < 0.01$, ***$P < 0.001$, compared between gene-specific silencing and the corresponding control of scrambled siRNA (dark and light bars of the same color).

same ChIP experiment but to detect a different site (~200 bp from TSS) in the S2R gene promoter that is low-ranked in SRE prediction. As seen in Fig 9C, the qPCR signal did not respond to U18 treatment or BRD2 expression, thus providing another negative control for the methodology in addition to IgG. It is known that BRD2 through its bromodomains binds the epigenomic mark H3K27Ac to facilitate transcriptional activation at select gene loci (19). We thus further corroborated the observed BRD2 occupancy of S2R promoter via ChIP-qPCR using a H3K27Ac-specific antibody (Fig 10). Consistently,

the data indicated that H3K27Ac enriched at the same S2R promoter site (~1,000 bp from TSS) where BRD2 occupied. In addition, we performed fluorescence imaging to illustrate protein subcellular distribution. As shown by Fig 11, GFP or mCherry alone (not in fusion with BRD2 or SREBP2Nterm) evenly dispersed throughout the whole cell, indicative of a nonspecific distribution pattern. However, both BRD2 and SREBP2Nterm, whether tagged with GFP or mCherry, were confined in the nucleus with high levels of overlap, verifying proper nuclear localization of these ectopically expressed proteins.

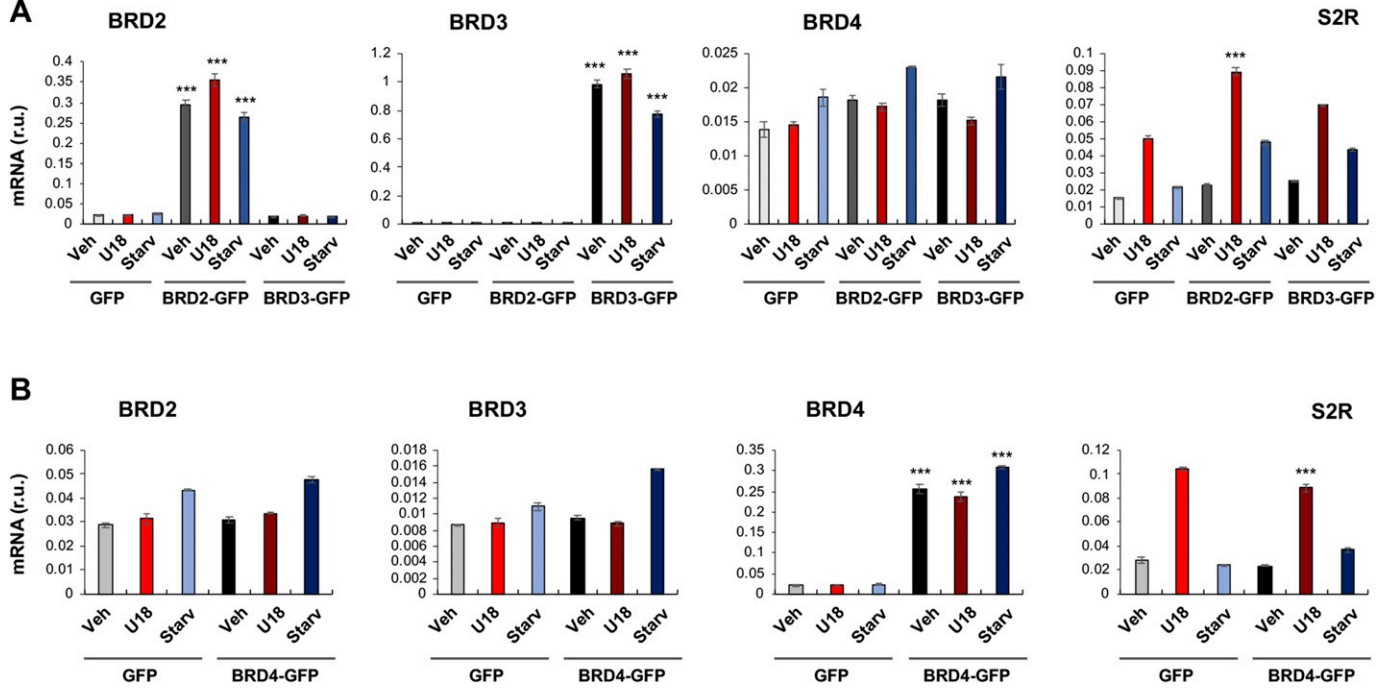

**Figure 3.   BRD2 but not BRD3 or BRD4 gain-of-function up-regulates sigma-2 receptor mRNA.**
**(A)** BRD2 and BRD3 gain-of function (plotted together). **(B)** BRD4 gain-of function. Experiments were performed as described for Fig 2. For gain-of-function, ARPE19 cells were transfected with a vector to express GFP (control) or BRD2-GFP or BRD3-GFP or BRD4-GFP. Data are presented as mean ± SD, n = 3 replicates. Each plot represents one of two similar experiments. Statistics: one-way ANOVA with Bonferroni post hoc test; ***$P < 0.001$, compared between BRD2 (or 3 or 4) overexpression and its corresponding GFP control (dark and light bars of the same color).

In aggregate, these results suggest that BRD2 up-regulates S2R expression not by increasing SREBP2 protein production, but rather, by forming a BRD2/SREBP2 complex that occupies the S2R gene's promoter to activate its transcription (see the schematic working model in Fig 12).

## Discussion

Cholesterol dysregulation leads to a myriad of pathological conditions (16). S2R was recently suggested as a novel player in cholesterol intracellular transport (3). To the best of our knowledge upon the preparation of this manuscript (17 *Preprint*), our report is the first to reveal BRD2-governed S2R expression. We differentiated that silencing BRD2 but not BRD4 (though widely deemed a master BET) or BRD3 effectively reduced S2R expression. Whereas pan-BETs inhibition blocked the transcription of both SREBPs, silencing SREBP2 but not SREBP1 repressed S2R expression. Furthermore, our data provided evidence for that BRD2 controls S2R transcription not by increasing the SREBP2 protein but by forming a BRD2/SREBP2 complex at the S2R gene promoter. Thus, our study suggests a previously unrecognized mechanism whereby the duo of BRD2/SREBP2 positively regulates S2R expression in response to the cholesterol level drop in the ER.

The finding of BET-dominated regulation of S2R expression is significant for the following reasons. (1) Epigenetics is crucial in cellular responses to extra- and intra-cellular environmental cues; however, little is known about regulations of S2R (23), whose expression is highly sensitive to cholesterol level perturbation. This knowledge gap likely stems from the fact that S2R is one of very few drug targets whose coding gene remained unknown until very recently (4). (2) S2R was implicated (via pharmacology) in neurological diseases (24) (e.g., Alzheimer's) (25), psychiatric disorders (6, 26), and cancers (27). In fact, S2R ligands have long been clinically used as antidepressants (e.g., haloperidol) (6). Moreover, high S2R abundance was found in tumor tissues and cells, as detected with labeled S2R ligands (7, 27) or unknowingly as TMEM97/MAC30 (28). As such, S2R is often targeted for cancer imaging (e.g., PET scanning) (7). More recently, it was reported that S2R (TMEM97) knockdown attenuated Niemann–Pick disease phenotypes in a mouse model, linking S2R to lysosomal cholesterol export (2). (3) S2R was found to re-locate to lysosomes when intracellular cholesterol levels dropped (3). It was thus speculated that S2R may interact with NPC1 modulating its cholesterol-exporting function in the lysosomal membrane (3). (4) Recent studies indicated that S2R co-localizes with LDLR, which via internalization carries esterified cholesterol into the cell (3); S2R knockdown impairs cholesterol (and LDLR) uptake. Together, these studies indicated the biological importance of S2R and its regulators and motivated our investigation into BETs-dictated regulation of S2R expression.

In this novel regulation, it is somewhat surprising that BRD2 rather than BRD4 is found to be the determinant BET. In contrast to BRD2 and BRD3, BRD4 has been intensively studied and shown to play a critical role in many crucial cellular processes and pathological conditions (8, 29). The BRD4 molecule (versus BRD2 and BRD3) has nearly doubled length which contains a unique C-terminal domain.

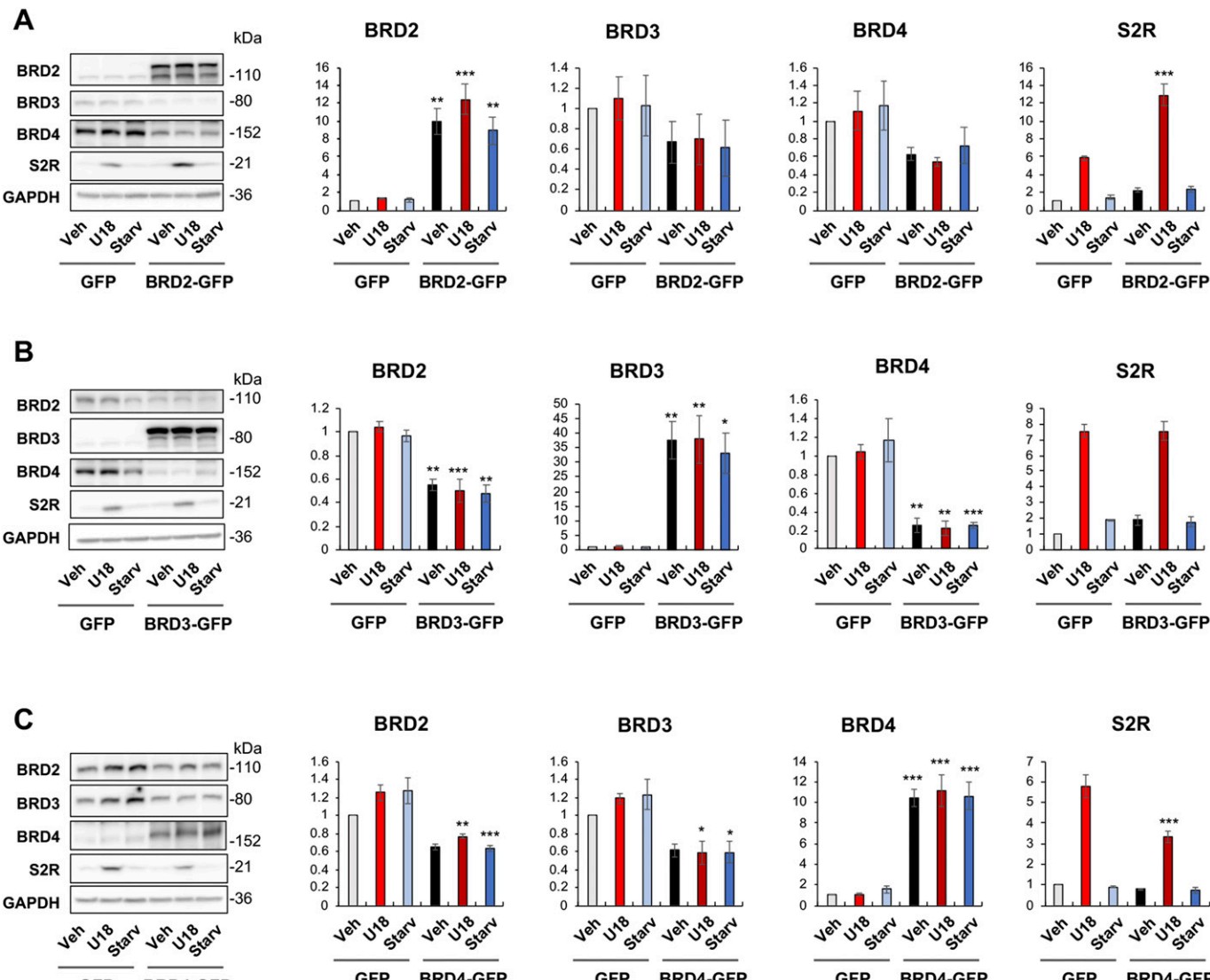

**Figure 4.   BRD2 but not BRD3 or BRD4 gain-of-function up-regulates sigma-2 receptor protein.**
**(A, B, C)** BRD2, BRD3, and BRD4 gain-of-function, respectively. Experiments and Western blot data quantification were performed as described for Fig 2. ARPE19 cells were transfected with a vector to express GFP (control) or BRD2-GFP or BRD3-GFP or BRD4-GFP. Statistics: one-way ANOVA with Bonferroni post hoc test; n = 4 independent repeat experiments; *$P < 0.05$, **$P < 0.01$, ***$P < 0.001$, compared between BRD2 (or 3 or 4) overexpression and its corresponding GFP control (dark and light bars of the same color).

From a functional/structural perspective, BRD4 is dubbed a "Swiss army knife" (30). Its two bromodomains "dock" the BRD4-organized regulatory complex (including TFs and cis-regulators) to specific bookmarked chromatin sites, with its C-terminal domain promoting transcriptional activation by interacting with the transcription elongation factor that in turn activates the RNA polymerase II (31). BRD4 was very recently found to possess intrinsic kinase (32) and acetyl transferase activities (33). However, in our specific experimental setting of U18-induced cytosolic cholesterol deprivation, it was BRD2 but not BRD4 that positively regulated S2R transcription. Inasmuch as BRD2 lacks a C-terminal domain and its bromodomain sequences are different from that of BRD4 (20), our results may implicate a BET mechanism distinct from that of BRD4. Given limited information about BRD2 functional mechanisms (31), future studies

are needed to elucidate the molecular workings that underlie BRD2-dominated regulation of S2R transcription.

To this end, the previous evidence for S2R being a target gene of SREBP2 (3) inspired our investigation that led to another novel finding, that is, BETs govern the transcription of both SREBP1 and SREBP2, the TFs key to fatty acid/cholesterol regulations. This BETs control over SREBP's transcription is exciting, given that BETs and SREBPs are both master regulators of vital cellular activities, yet their relationship was previously unknown in the context of cholesterol homeostasis where SREBPs are critically important. Thanks to the recent discovery of inhibitors selective for BETs (19, 34), important BET functions have been recently identified. BETs inhibition was initially shown to be highly effective in blocking transcription programs of inflammation (34) and oncogenic proliferation (35).

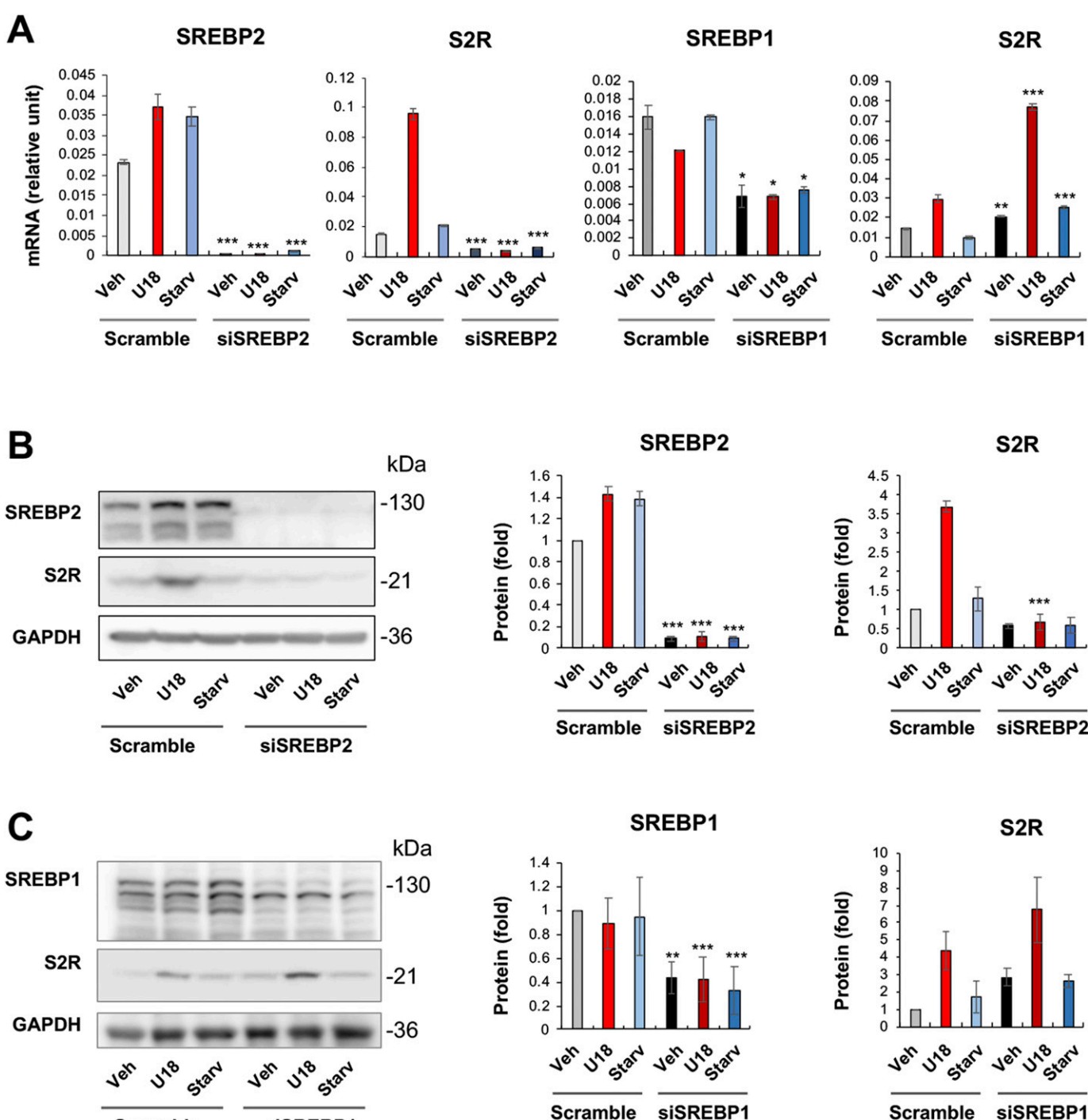

**Figure 5. SREBP2 but not SREBP1 silencing reduces sigma-2 receptor mRNA and protein expression.**
**(A)** mRNA levels (qRT-PCR). Quantification: Mean ± SD, n = 3 replicates. Each plot represents one of two similar experiments. **(B, C)** Protein levels (Western blot). Experiments with ARPE19 cells and data quantification were performed as described for Fig 2. Statistics: one-way ANOVA with Bonferroni post hoc test; n = 4 independent repeat experiments; *$P < 0.05$, **$P < 0.01$, ***$P < 0.001$, compared between gene-specific silencing and its corresponding control of scrambled siRNA (dark and light bars of the same color).

The importance of BET biology was then extended to stem cell differentiation, hematopoiesis, synaptic plasticity (13), and recently, also adipogenesis (10, 36). Most recently, BRD4 was found to regulate intra-nuclear/cellular processes as well, such as chromatin

architectural remodeling (37) and autophagy (12). However, there is a dearth of information on a role for BETs in regulating SREBPs. The closest relevance is a report on BET-associated super-enhancers formed during adipogenesis (10). While SREBP1 (but not SREBP2)

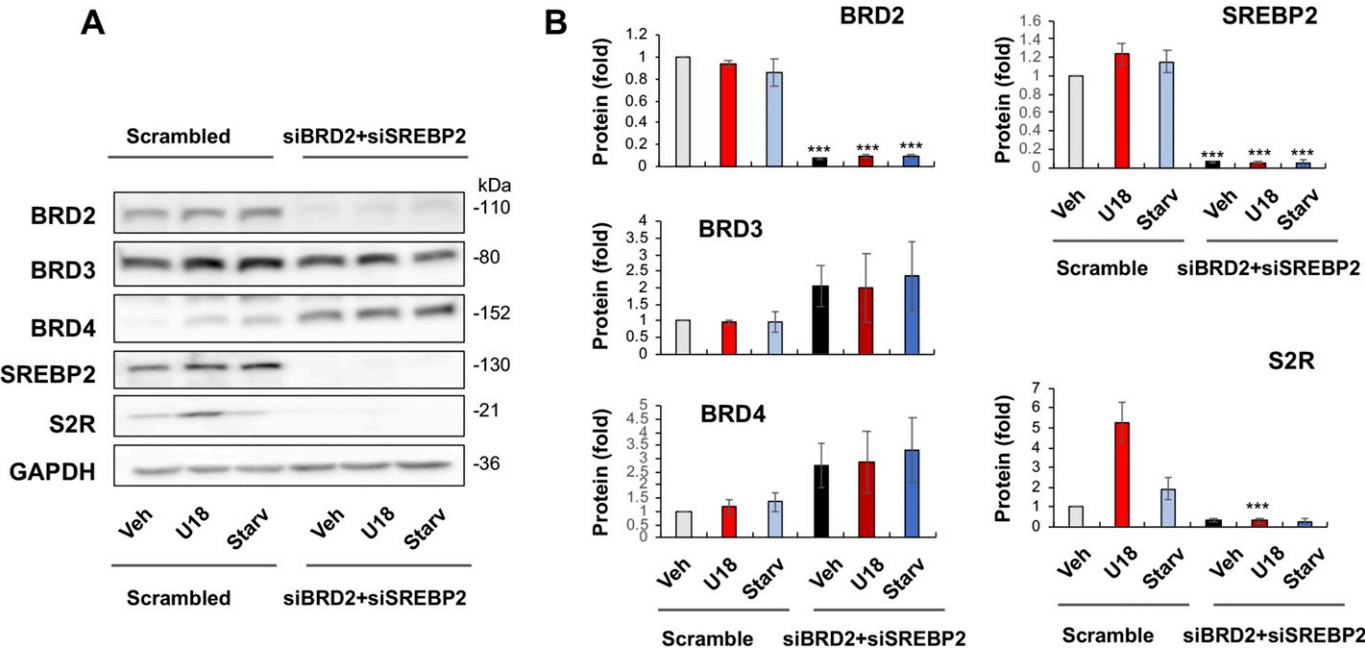

**Figure 6. BRD2 and SREBP2 double silencing diminishes sigma-2 receptor expression.**
**(A)** Representative Western blots. **(B)** Quantified data. Experiments and Western blot data quantification were performed as described for Fig 2. BRD2-specific and SREBP2-specific siRNAs were used separately or in combination for gene silencing. Statistics: one-way ANOVA with Bonferroni post hoc test; n = 4 independent repeat experiments; ***$P$ < 0.001, compared between gene-specific silencing and the scrambled siRNA control (dark and light bars of the same color).

was on the list of JQ1-regulated genes derived from RNAseq data, the relationship between BETs and SREBP1 was not specifically examined. Increased HDL cholesterol or decreased LDL cholesterol was observed in animal plasma after treating with a BETs inhibitor (38, 39). However, little is known about specific BET-dominated cholesterol regulatory pathways, underscoring the importance of investigating BETs regulations of SREBPs and their downstream effector genes.

SREBPs could be reasonably categorized as master TFs. There are total more than 1,000 TFs, but only a limited number of them are deemed master TFs. These master TFs often direct transcription programs that define a cell type or cell state (8). As such, they are sensitive to extra- and intracellular environmental perturbations, and their expression levels and activities are key to cell state changes and associated disease conditions (40). Recently, master TFs were found to co-localize with BRD4 in the genomic landscape thereby playing a critical role in inflammatory (e.g., NFκB) (40), proliferative (c-Myc) (35, 41), or immunological (T-bet) processes. Another prominent feature of master TFs is that they potently co-activate the transcription of specific sets of genes, by forming a complex with BETs to promote not only target gene expression but also their own transcription (8). In consonance with this scenario, our ChIP-qPCR data showed BRD2 occupancy at the SREBP2 gene promoter (Fig S6); indeed, blocking BETs with JQ1 reduced transcripts of both SREBP2 and SREBP1 genes (Fig 7). Based on these criteria, SREBP2 and SREBP1 appear to be master TFs. SREBP1 and SREBP2 differentially regulate fatty acid and cholesterol pathways (though with possible crosstalk). This may explain our observation that SREBP2 but not SREBP1 positively regulated S2R expression upon cholesterol deprivation. Of note, serum starvation markedly stimulated the expression of both SREBP2 and SREBP1, likely because serum contains both cholesterol and fatty acids. Consistently, pan-BETs inhibition with JQ1 averted up-regulation of both SREBP1 and SREBP2 mRNAs stimulated by serum starvation. In addition, JQ1 also reduced mRNAs of SREBP target genes tested herein including not only S2R, but NPC1, NPC2, and LXRs as well (Fig S9). As such, BETs together with SREBPs may govern transcription programs of the cholesterol/fatty acid pathways. For this possibility, future RNAseq studies are needed to provide more comprehensive evidence.

Furthermore, our results suggest that in the condition of cholesterol deprivation the BRD2/SREBP2 duo accounted for the up-regulation of S2R. Evidence includes the following: (1) U18 treatment dramatically up-regulated S2R and SREBP2 but not SREBP1 expression. (2) Silencing SREBP2 but not SREBP1 reduced S2R mRNA and protein. (3) BRD2 co-immunoprecipitated with the SREBP2 N-terminal TF domain. (4) ChIP-qPCR assays using a BRD2 antibody suggested that BRD2/SREBP2 co-occupied S2R promoter regions, and this co-occupancy was enhanced by U18-induced cholesterol deprivation and BRD2 overexpression. This novel result is consistent with the well-documented SREBP2 functional mechanism; namely, SREBP2 in an inactive state resides in the ER membrane, but upon drop of cytosolic/ER cholesterol levels, SREBP2 is transported to Golgi and cleaved into two half molecules (3). Whereas the C-terminal half stays in the cytosol, the N-terminal half enters the nucleus acting as a TF for the expression of genes involved in cholesterol metabolism and transport (22). Taken together, the pockets of new information obtained herein and that from the literature form a coherent picture of

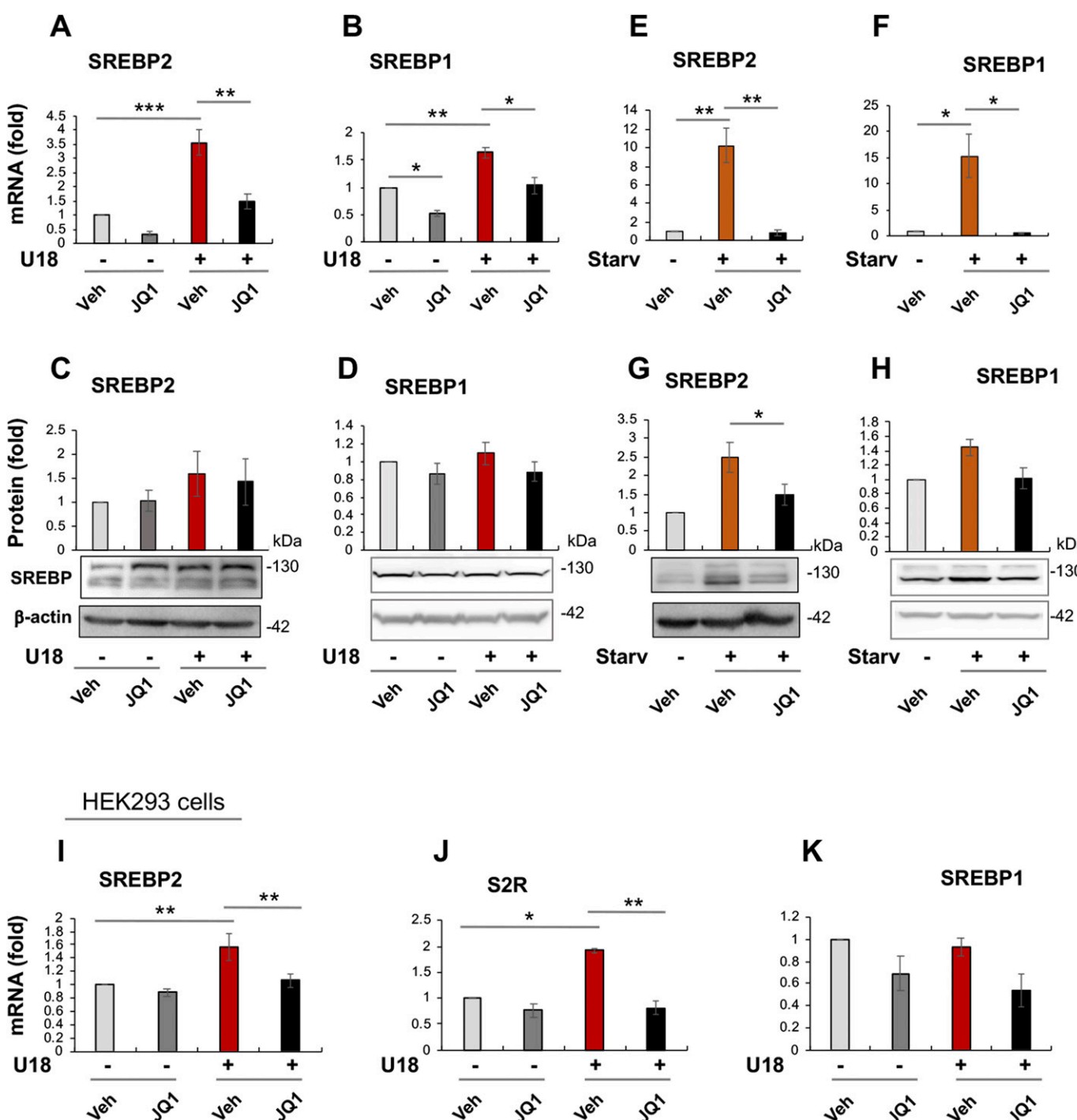

**Figure 7. BET inhibition suppresses the transcription of SREBPs.**
**(A, B)** Effect of JQ1 on SREBP2 and SREBP1 mRNA levels in U18-treated ARPE19 cells. **(C, D)** Effect of JQ1 on SREBP2 and SREBP1 protein levels in U18-treated ARPE19 cells.
**(E, F)** Effect of JQ1 on SREBP2 and SREBP1 mRNA levels in starved ARPE19 cells. **(G, H)** Effect of JQ1 on SREBP2 and SREBP1 protein levels in starved ARPE19 cells. **(I, J, K)** Effect of JQ1 on SREBP2, sigma-2 receptor, and SREBP1 mRNA levels, respectively, in HEK293 cells. Experimental procedures and data quantification were performed as described for Fig 1. Statistics: one-way ANOVA with Bonferroni post hoc test; n = 3 (qRT-PCR) or 4 (Western blot) independent repeat experiments; *$P < 0.05$, **$P < 0.01$, ***$P < 0.001$; for simplicity, nonsignificant comparison is not labeled.
Source data are available for this figure.

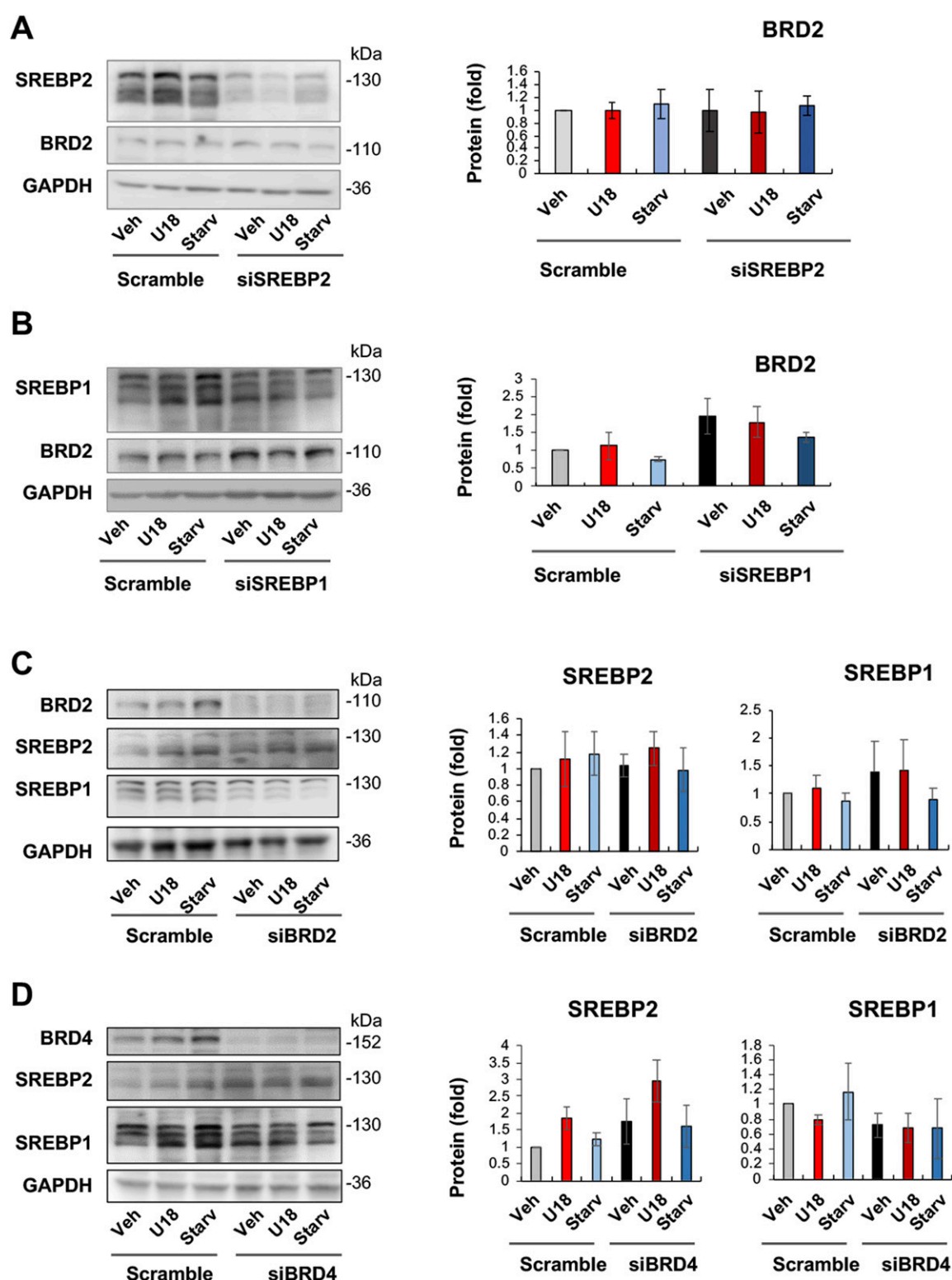

**Figure 8. SREBP2 silencing does not alter BRD2 protein levels.**
**(A)** Effect of SREBP2 silencing on BRD2 protein levels. **(B)** Effect of SREBP1 silencing on BRD2 protein levels. **(C)** Effect of BRD2 silencing on SREBP2 and SREBP1 protein levels. **(D)** Effect of BRD4 silencing on SREBP2 and SREBP1 protein levels. ARPE19 cells were transfected with a scrambled or specific siRNA, treated for 24 h with U18 or starvation, and then harvested for Western blot assay. Data (mean ± SEM) were quantified as described for Fig 1. Statistics: one-way ANOVA with Bonferroni post hoc test; n = 4 independent repeat experiments. Nonsignificant difference is not labeled.

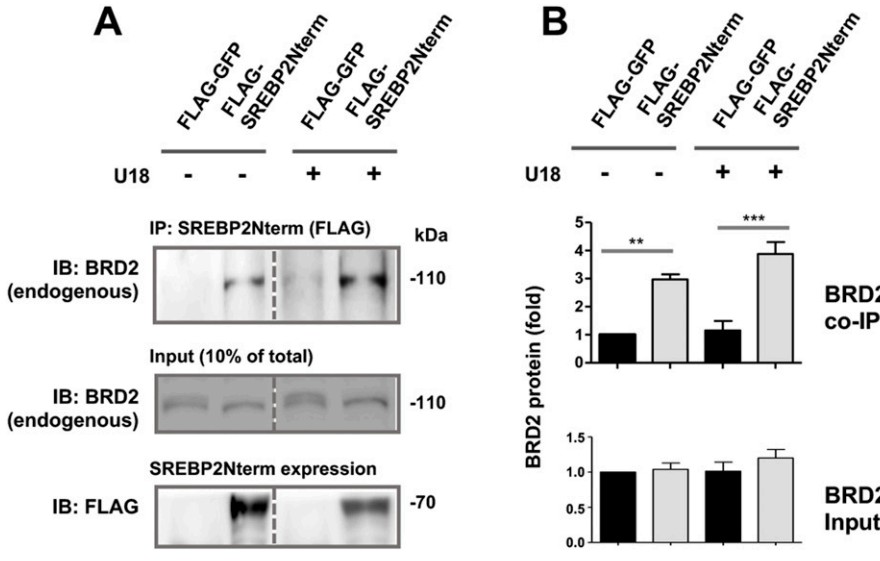

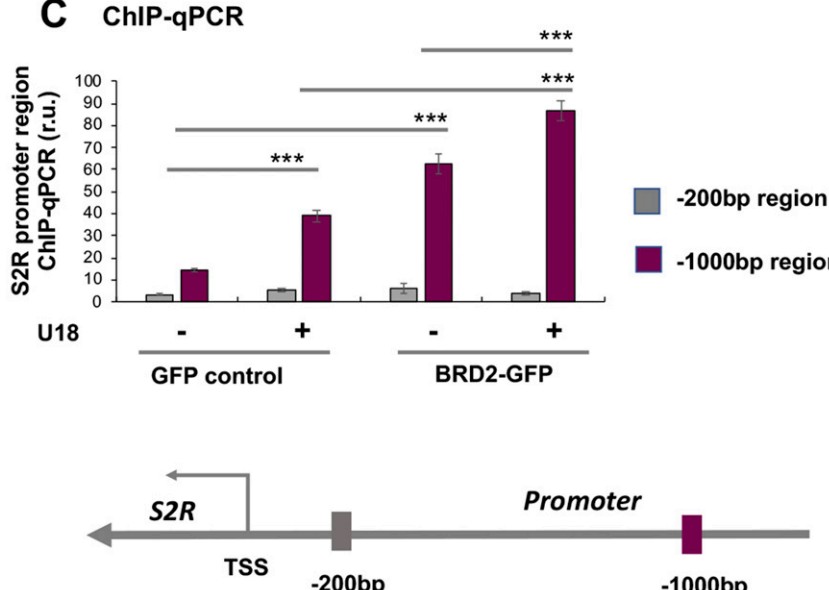

**Figure 9. Endogenous BRD2 co-immunoprecipitates with the SREBP2 N-terminal half molecule and sigma-2 receptor gene promoter DNA.**
**(A, B)** Co-immuoprecipitation (Co-IP). ARPE19 cells were transfected with a vector to express FLAG-GFP (control) or FLAG-SREBP2Nterm (the transcriptionally active N-terminal half molecule). IP was performed with an anti-FLAG antibody and immunoblotting (IB) with an antibody against endogenous BRD2. Quantification: mean ± SEM; n = 3 independent repeat experiments. The value of basal control (FLAG-GFP, no U18) was used to normalize data. Statistics: one-way ANOVA with Bonferroni post hoc test; **$P < 0.01$, ***$P < 0.001$. **(C)** ChIP-qPCR. Cells were transfected with a vector to express GFP (control) or BRD2-GFP. ChIP was performed using an antibody against endogenous BRD2. qPCR was performed to detect a software-predicted SREBP-binding region (~1,000 bp from the Transcription Start Site, TSS) in the sigma-2 receptor gene promoter and a relatively unrelated region (~200 bp) for negative control. The value from the basal control was used to normalize the ChIP-qPCR data. Quantification: mean ± SD; n = 3 replicates. Statistics: one-way ANOVA with Bonferroni post hoc test; ***$P < 0.001$.
Source data are available for this figure.

BRD2/SREBP2-dictated S2R transcription in response to cytosolic cholesterol perturbation.

## Conclusions

We present here the first mechanistic study on the BRD2 control over the transcription of S2R, a recently unveiled player in cholesterol homeostasis. Our results reveal a novel regulation, whereby BRD2 forms a complex with master TF SREBP2 at the S2R gene promoter activating its transcription. Further investigation may shed new lights on BRD2-dominated transcription programs that sense cholesterol perturbation. Along this line, studies on BETs and S2R,

both targets of increasing clinical (or trial) drugs (6, 14, 29, 42, 43), may synergize interventional opportunities for cholesterol-associated pathological conditions.

# Materials and Methods

## Major materials

JQ1 was purchased from Apexbio (A1910). U18666A was from Sigma-Aldrich (662015). Filipin complex was from Sigma-Aldrich (F9765). ARPE19 cells and HEK293 cells were obtained from American Type Culture Collection.

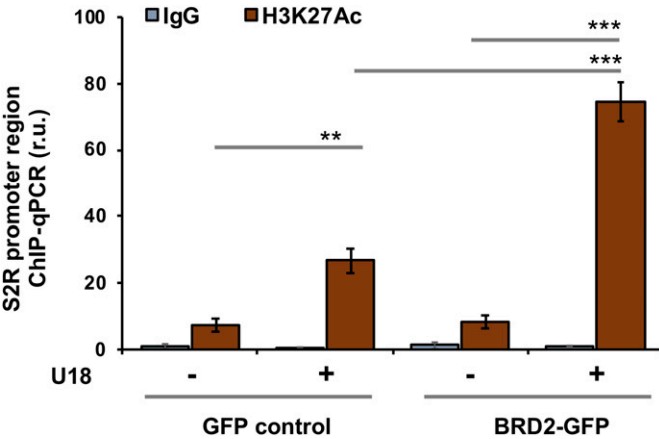

**Figure 10. H3K27Ac co-immunoprecipitates the sigma-2 receptor gene promoter DNA.**
Cells were transfected with a vector to express GFP (control) or BRD2-GFP. ChIP was performed using an antibody against H3K27Ac. qPCR was performed to detect a software-predicted SREBP-binding region (~1,000 bp from the Transcription Start Site, the same as in Fig 9C) in the sigma-2 receptor gene promoter. The value from the basal control was used to normalize the ChIP-qPCR data. Quantification: mean ± SD; n = 3 replicates. Statistics: one-way ANOVA with Bonferroni post hoc test; ***$P < 0.001$, **$P < 0.01$.

Scrambled and BRD2-, BRD4-, or SREBP1-specific siRNAs were from Thermo Fisher Scientific (scrambled: AM4635; BRD2: AM16708, ID-118266; BRD4: 4457298, ID-s23902; SREBP1: AM51331, ID-5140). Lipofectamine 3000 was from Thermo Fisher Scientific (L3000008).

### Cell culture and treatment

ARPE19 cells were cultured in the DMEM/F12 medium (11320082; Thermo Fisher Scientific) supplemented with 10% FBS and penicillin/streptomycin (5140163; Thermo Fisher Scientific) at 37°C in a humidified atmosphere with 5% $CO_2$. HEK293 cells were maintained in DMEM (10569010; Thermo Fisher Scientific) supplemented with 10% FBS and penicillin/streptomycin. To induce cytosolic cholesterol deprivation, ARPE19 cells were seeded in six-well plates at $3 \times 10^5$ cells/well and cultured for 24 h, U18666A was then added (final 5 $\mu$M) and incubated for another 24 h. In some experiments, JQ1 was added (1 $\mu$M) together with U18666A. For starvation experiments, the culture medium containing 10% FBS was changed to that without FBS in which cells were cultured for 24 h. Cells were then collected for various analyses. For cholesterol staining, the cells were washed 3× with PBS before fixing with 3% paraformaldehyde for 1 h. The reaction was stopped with glycine (1.5 mg/ml) and the cells were then stained for 2 h at room temperature in the filipin working solution (0.05 mg/ml in PBS with 10% FBS). Images were taken with the Nikon fluorescence microscope using a UV filter set (340–380 nm excitation).

### RNA isolation, reverse transcription, and quantitative real-time PCR (qRT-PCR)

Total RNA was isolated and purified using Trizol Reagent (15596026; Thermo Fisher Scientific) following the manufacturer's instruction. RNA was reverse-transcribed using the High-Capacity cDNA Reverse

Transcription kit (4368814; Thermo Fisher Scientific). cDNA of 1 $\mu$l from 20 $\mu$l reaction volume was amplified by real-time quantitative PCR (Applied Biosystems Quant Studio 3; Thermo Fisher Scientific) with Perfecta SYBR Green Master Fast Mix (101414-286; VWR) (11). Relative gene expression was determined by the $2^{-\Delta\Delta Ct}$ method, normalized to GAPDH, and presented as relative mRNA levels. qPCR analyses were performed in triplicate. Experiments were repeated at least twice. Primers are listed in Table S1.

### Preparation of lentivectors for shRNA expression

The pLKO.1-puro empty vector was purchased from Addgene (#8453). A scrambled shRNA control and gene-specific shRNAs were designed through RNAi Central (http://cancan.cshl.edu/RNAi_central/step2.cgi). The corresponding oligonucleotides (ordered from Thermo Fisher Scientific) were annealed (95°C–25°C, 0.1°C/s) and cloned into the pLKO.1-puro vector, followed by confirmatory sequencing at the Ohio State University facility. The shRNA sequences (of final siRNA products) are listed in Table S2. For lentivirus packaging, lentivector plasmids were transfected into HEK293T cells together with packaging and envelope plasmids (psPAX2 and pMD2.G) using Lipofectamine 3000 (L3000008; Thermo Fisher Scientific). 3 d after transfection, the medium was passed through a filter of 0.45 $\mu$m pore size and then used for the transduction of ARPE19 cells. After 48 h of infection, the cells were selected with 1 $\mu$g/ml of puromycin (A1113803; Thermo Fisher Scientific) for 5–10 d.

### Plasmid and siRNA transfection

Plasmids were transfected via co-incubation with Lipofectamine 3000 for 24 h in the recipient cell culture following the manufacturer's instructions. The medium was then replaced with fresh DMEM/F12 for the cells to recover (24 h) before their further use in various analyses.

For siRNA transfection, ARPE19 cells were cultured to 80% confluency in the DMEM/F12 medium containing 10% FBS in six-well plates, and then added with a scrambled or BRD2-, BRD4-, or SREBP1-specific siRNA (sequence information available at the manufacturer; Thermo Fisher Scientific). The cells were transfected overnight using the Lipofectamine RNAi Max transfection reagent (13778150; Thermo Fisher Scientific), and then recovered in the DMEM/F12 culture medium for 24 h before further experimental use.

### Western blotting

Western blot analysis was performed following our published protocol (11) with minor modification. Briefly, cells were lysed with the Pierce RIPA lysis buffer (89901; Thermo Fisher Scientific) containing Halt Protease Inhibitor Cocktail (87785; Thermo Fisher Scientific). Total protein concentration was determined using the DC Protein Assay Kit (5000111; Bio-Rad). The cell lysates were solubilized in Pierce Lane Marker Non-Reducing Sample Buffer (39001; Thermo Fisher Scientific) and heated at 95°C for 10 min after SDS–PAGE and Western blotting. The information for the antibodies used are available in Table S3. Specific protein bands on Western blots

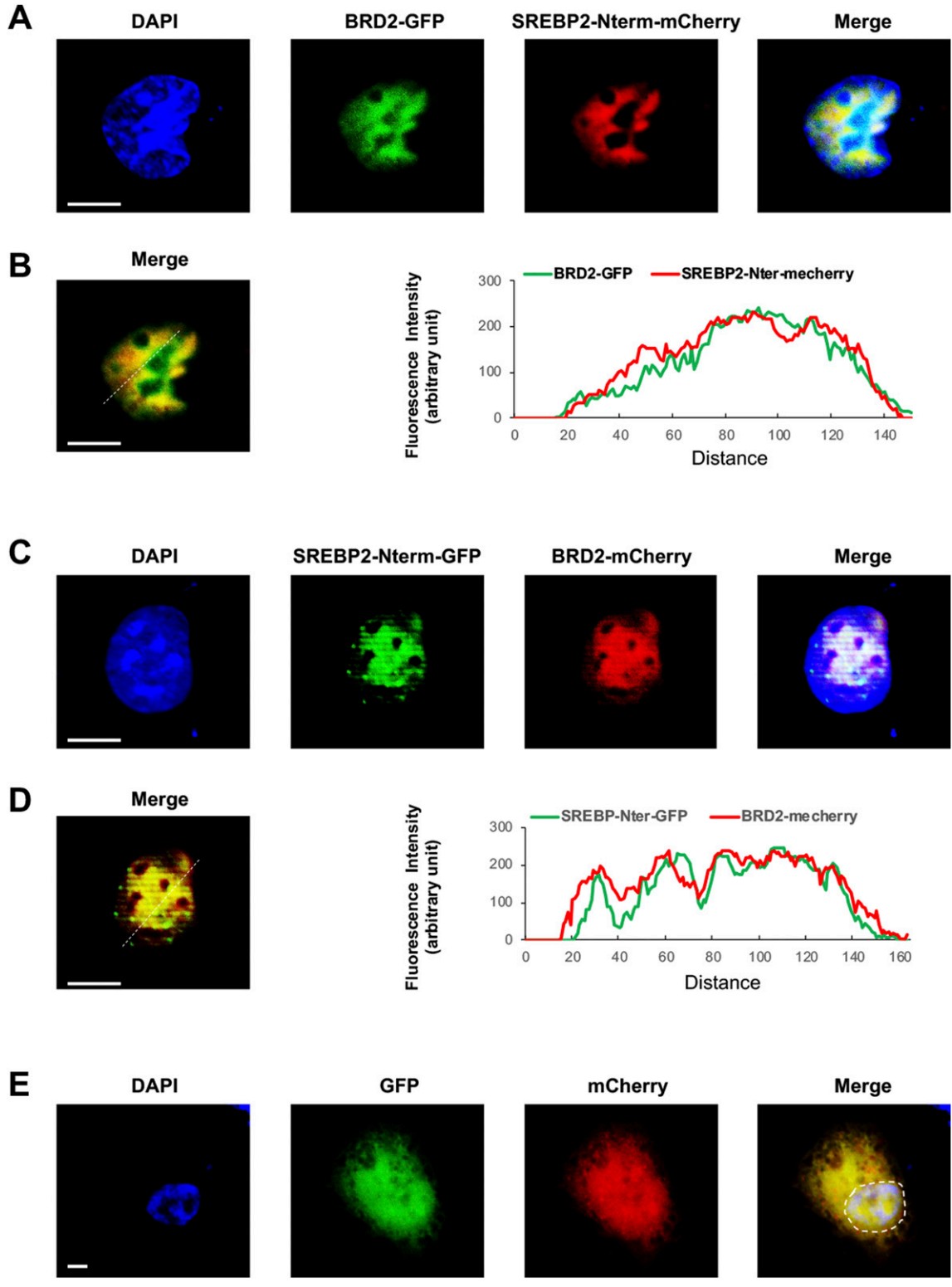

**Figure 11. Localization of BRD2 and SREBP2 in the nucleus.**
To confirm the nuclear localization of overexpressed proteins, ARPE19 cells were transfected with a vector to express SREBP2Nterm or BRD2, each tagged with GFP or mCherry. Confocal microscopy was performed for imaging. Co-localization of these two proteins is shown by merged images and also by their similar profiles of fluorescence intensity (along the dashed line). By contrast, GFP or mCherry alone is evenly distributed throughout the whole cell. Scale bar: 5 μm. **(A, B, C, D, E)** show different protein tagging; (B, D) show nuclear fluorescence distribution profiles (along the dashed line); (E) shows nonspecific distribution of GFP or mCherry alone (not in fusion).

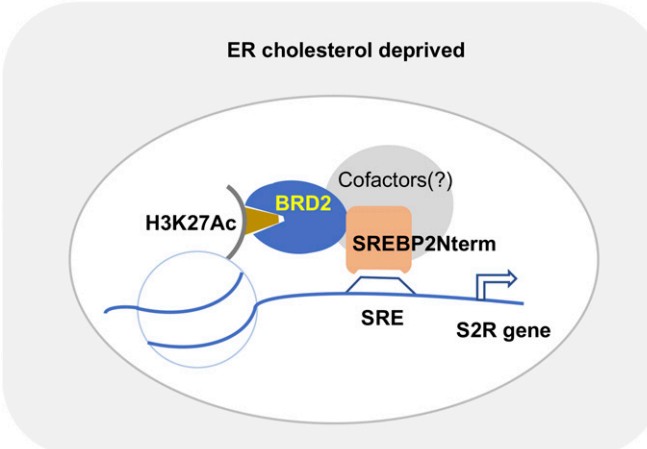

**Figure 12. Schematic model of BRD2/SREBP2 cooperative regulation of sigma-2 receptor (S2R) transcription.**
Treatment of ARPE19 cells with U18, a NPC1 inhibitor, blocks cholesterol export from lysosomes thereby generating an intracellular environment where ER cholesterol is deprived. Sensing this cholesterol level change, SREBP2 is activated. Its inhibitory C-terminal half is cleaved off and the N-terminal half (SREBP2Nterm) is able to enter the nucleus to bind an SRE in the S2R gene promoter. At the same genomic loci, BRD2 is enriched by binding to acetylated histone while interacting with the SREBP2Nterm, directly or indirectly through cofactors (to be determined). This complex facilitates activation of S2R gene expression via the transcription machinery.

were quantified by the ImageJ 64 software (https://imagej.nih.gov/ij/) using Gel analyzer script. Densitometry data were normalized to loading control (GAPDH or $\beta$-actin) and then to a basal condition (e.g., vehicle and/or scrambled sequence siRNA).

### Co-immunoprecipitation (co-IP)

pEGFP-N1-FLAG (empty vector) (60360; Addgene) and pFLAG-SREBP2Nt (N-terminal transcriptionally active domain, amino acids 1–482) (26807; Addgene) were used to transfect cells (HEK293). For co-IP, cells were lysed on ice for 30 min in Pierce IP Lysis Buffer (87788; Thermo Fisher Scientific) containing Halt Protease Inhibitor Cocktail (87785; Thermo Fisher Scientific), and then centrifuged at 13,200$g$ for 15 min at 4°C. The supernatant was incubated with 50 $\mu$l of Pierce Anti-DYKDDDDK Magnetic Agarose beads (A36797; Thermo Fisher Scientific) at 4°C overnight. The beads were washed 3× with cold PBS buffer and then incubated in 0.1 M glycine (pH 2.8) for 10 min at room temperature with frequent vortex to elute the immunoprecipitates. The eluate was neutralized with 1M Tris–HCl, pH 8.5 (15 $\mu$l per 100 $\mu$l eluate), and briefly heated at 95°C before its use for SDS–PAGE and Western blot analysis.

### Chromatin immunoprecipitation (ChIP) analysis

ChIP analysis was performed by using the Pierce Magnetic ChIP kit (26157; Thermo Fisher Scientific) and following the manufacturer's manual. Briefly, formaldehyde (final concentration 1%) was incubated with the ARPE19 cell culture to cross-link protein with DNA for 10 min, and the reaction was then quenched with a glycine solution

for 5 min. Cells were washed with ice-cold PBS, lysed in the Membrane Extraction buffer, and centrifuged at 3,000$g$ for 5 min to collect the nuclei. The nuclear pellets were re-suspended in 200 $\mu$l of MNase Digestion Buffer Working Solution and then digested by incubation with MNase at 37°C for 15 min. The reaction was terminated in MNase Stop Solution. The nuclei were recovered by centrifugation at 9,000$g$ for 5 min, re-suspended in IP Dilution Buffer, and then sonicated (four 5-s pulses at 20 W for 2 × 10$^6$ cells) to break the nuclear membrane. After centrifugation at 9,000$g$ for 5 min, the supernatant was collected and incubated overnight with a BRD2 antibody (Table S3) or IgG control (5 $\mu$g antibody per reaction). ChIP-grade Protein A/G Magnetic beads were added and incubated overnight at 4°C on mixing. The beads were collected and washed sequentially with IP Wash Buffer-1, IP Wash Buffer-2, and then resuspended in the elution buffer. The protein-DNA crosslink was reversed with 5 M NaCl followed by RNA and protein digestion with RNAse A and Proteinase K. The DNA pull-down was purified with DNA Clean-Up Column and used for qPCR. Primers used for detection of S2R and SREBP promoter regions that contain predicted SREBP-binding sterol-response consensus elements are listed in Table S4. For each qPCR assay, triplicate samples were used, and data were normalized to respective input samples. For BRD2 gain of function, we selected a BRD2-expressing stable ARPE19 cell line using a lentivector constructed based on an empty vector (19319; Addgene).

### Statistical analysis

Repeat experiments were performed on different (n ≥ 3) occasions. Results were plotted as mean ± SEM unless otherwise specified. Statistical significance (set at $P < 0.05$) was determined by one-way ANOVA with Bonferroni post hoc test for multigroup comparison or two-tailed paired $t$ test for two-group comparison (GraphPad Prism 7). Significance is indicated as *$P < 0.05$, **$P < 0.01$, or ***$P < 0.001$; no significance is labeled as "ns" or not labeled, as specified in each figure legend.

# Supplementary Information

# Acknowledgements

This work was supported by National Institutes of Health (NIH) R01 grants HL133665 and EY029809 to L-W Guo, HL143469 and HL129785 to KC Kent and L-W Guo, and an American Heart Association pre-doctoral award 17PRE33670865 (to MX Zhang).

### Author Contributions

H Shen: data curation, formal analysis, investigation, and methodology.
J Li: data curation, investigation, and methodology.
X Xie: data curation and methodology.
H Yang: methodology.

M Zhang: investigation.

B Wang: formal analysis and investigation.

KC Kent: funding acquisition and project administration.

J Plutzky: resources, funding acquisition, and validation.

L-W Guo: conceptualization, formal analysis, manuscript writing, and funding acquisition.

## Conflict of Interest Statement

The authors declare that they have no conflict of interest.

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
