## [Reviewer comments · Life Science Alliance]

Life Science Alliance

BRD2 regulation of sigma-2 receptor upon cholesterol deprivation

Hongtao Shen, jing Li, Xiujie Xie, Huan Yang, Mengxue Zhang, Bowen Wang, Craig Kent, Jorge Plutzky, and Lian-Wang Guo

DOI: <https://doi.org/10.26508/lsa.201900540>

Corresponding author(s): Lian-Wang Guo, University of Virginia

Review Timeline:

Submission Date:	2019-09-07
Editorial Decision:	2019-10-08
Appeal Received:	2019-10-16
Editorial Decision:	2019-10-16
Revision Received:	2020-10-20
Editorial Decision:	2020-10-26
Revision Received:	2020-10-28
Editorial Decision:	2020-11-02
Revision Received:	2020-11-10
Accepted:	2020-11-11

Scientific Editor: Shachi Bhatt

Transaction Report:

October 8, 2019

Re: Life Science Alliance manuscript #LSA-2019-00540

Dr. Lian-Wang Guo
The Ohio State University
473 West 12th Ave
Columbus Columbus

Dear Dr. Guo,

Thank you for submitting your manuscript entitled "BRD2 regulation of sigma-2 receptor expression upon cytosolic cholesterol deprivation". The manuscript has been evaluated by expert reviewers, whose reports are appended below.

As you will see, the reviewers appreciate the topic of your work. However, they think that your conclusions are currently not sufficiently supported by the data provided. While the concerns of reviewer #1 could get addressed in a normal revision period, reviewer #2 points out that many of the data presented are not very robust at this stage and that it would be a major undertaking to elevate the robustness and to thus allow publication. We therefore concluded that we have to return your manuscript to you to allow more rapid publication elsewhere.

That being said, reviewer #2 is providing a lot of guidance to indicate which parts of the work need further support by alternative experiments or by repeat experiments. Should you be willing to invest the time and effort that would be required to address the reviewer concerns, we'd be happy to take a look at a fully revised version in the future. If you would like to resubmit this work to Life Science Alliance, please contact the journal office to discuss an appeal of this decision or you may submit an appeal directly through our manuscript submission system. Please note that priority and novelty would be reassessed at resubmission.

Regardless of how you choose to proceed, we hope that the comments below will prove constructive as your work progresses.

Thank you for thinking of Life Science Alliance as an appropriate place to publish your work.

Sincerely,

Reviewer #1 (Comments to the Authors (Required)):

The authors investigated the transcriptional regulation of sigma-2 receptor (S2R) by the BET family protein BRD2. They showed that S2R transcript and protein levels increase upon cholesterol deprivation by an inhibitor of Niemann-Pick disease, type C1 (NPC1), a lysosomal cholesterol transporter and starvation. This S2R upregulation is abolished by the BET inhibitor JQ1 and BRD2 knockdown, while knockdown of other BET proteins, BRD3 and BRD4, has no effect on S2R levels. BRD2 forms a complex with SREBP2 and binds to the promoter region of S2R gene, which provide a mechanistic insight into the S2R upregulation upon treatment of the NPC1 inhibitor U18666A and starvation. Consistently, SREBP2 knockdown attenuates S2R expression in cells treated with U18666A and under starvation.

The results are clear and the conclusion is supported by the data. Their findings that BRD2 responses cholesterol deprivation are novel and potentially advance our understanding of the regulation of cholesterol homeostasis. I have a couple of comments to strengthen the manuscript.

1. It is good to show that BRD2 does not bind to the further distal region of the S2R gene promoter.
2. They should conduct ChIP assay to show SREBP2 is recruited to the promoter region of S2R gene.
3. They should examine the interaction between BRD2 and the full-length or endogenous SREBP2.
4. U18666A should be U18666A.

Reviewer #2 (Comments to the Authors (Required)):

"BRD2 regulation of sigma-2 receptor expression upon cytosolic cholesterol deprivation".

SUMMARY

The authors discovered an epigenetic circuit not previously reported, which is triggered by fluctuations in cytoplasmic cholesterol levels. They report the new role for BET chromatin remodelers in the regulation of the expression of mRNA and protein of S2R, involved in the control of cholesterol homeostasis. BRD2 binds the transcription factor SREBP2 and both localize at the S2R gene promoter promoting its transcription. The emerging role of S2R as a regulator of intracellular cholesterol levels in physiology and pathology, make important the understanding of the transcriptional/epigenetic programs controlling its expression, which represent a potential pharmacological target for the treatment of cholesterol-related dysfunctions and tumors. However, although the manuscript flow can be improved, I see major and minor points that need to be addressed for this manuscript to be published.

MAJOR POINTS

- 1) The authors show the effect of cholesterol deprivation from the cytoplasm (might be better to state from ER) on the induction of S2R expression. There are two gaps here that remain unresolved: i) what is the sensor that detects the accumulation of cholesterol in lysosomes, or the deprivation from ER (SCAP for the ER?) and activates the epigenetic/transcriptional circuit resulting in S2R expression; ii) what is the biological meaning of S2R upregulation in ER cholesterol deprivation. They do not show any effect of the induction of the BRD2 / SREBP2 / S2R axis on uptake, total levels and on the distribution of intracellular cholesterol or LDLR. It is clear that this is a paper on the transcriptional regulation of S2R, but adding further details about the two mentioned

tasks, or at least discuss it better would be an added value for the paper.

2) The whole study was carried out by silencing the components of the disclosed circuit and no over-expression or rescue experiment was conducted.

3) Serum starvation is not an efficient way for total cholesterol deprivation. Playing with complete medium, a cholesterol-free medium, in combination with HMG CoA reductase and U18 inhibitors would be useful to get selective ER deprivation, accumulation in lysosomes or total cholesterol abrogation.

4) All experiments in blocking cholesterol output from lysosomes were conducted with the use of U18. It would be advisable to silencing NPC's transporters or use NPC1 mutant cells to see if phenocopy the drug effect.

5) U18 treatment of SCAP silenced cells might help in understanding the circuit better.

6) The quality of Co-IPs experiments is not satisfying and in the supplementary information. These represent the most important data of the paper.

7) The authors didn't dig in to the intrinsic complexity of the epigenetic circuit. As in the case of Figure 3F in which the role of BRD4 in the regulation of S2R expression is not well understood. An important experiment that would be the overexpression (OE) of BRD2, 3 and 4 in absence of cholesterol deprivation. The BRD2-OE might indicate its sufficiency for the expression of S2R. Another key experiment would be OE of BRD4 in combination with BRD2 silencing. Furthermore, the epistatic experiment of the double silencing of BRD2 and 4 underlines: i) a role for BRD2 in the expression of S2R; ii) a role of BRD4 in the expression of S2R through induction of BRD2 expression; iii) a role for BRD4, independent of BRD2, in the repression of S2R. In a paper studying the players involved in the regulation of S2R expression, the role of BRD4 in the repression of S2R cannot be neglected.

8) A figure with a graphical abstract would support the readers. Authors should provide a model of their findings.

9) In my opinion, this paper is not of satisfying quality, some experiments do not have proper controls and some blots and images are of very limited representative significance. The data are very preliminary and some of the Western blots are not of satisfying quality. In terms of scientific impact, the authors claim to have found a transcription factor complex that regulates the expression of S2R when lysosomal cholesterol levels are perturbed. No washout or rescue experiments to show reversibility of the phenotype have been performed throughout the paper. I believe that to carry out all the aforementioned experiments it is not feasible in a standard time of a revision, and at present the paper is in a preliminary state for publication.

MINOR POINTS

1) In Figure 1D the blot is of not satisfying quality for S2R. In fact, going by that blot, under JQ1 treatment results in a complete loss of S2R protein levels. I find it fascinating that the authors claim this is non-significant in their bar graphs. How did they quantify this blot?

2) In Figure 3 please try overexpression of the different BRDs and check the effect on S2R mRNA and protein expression;

3) In Figure 4, the levels of knock down of SREBP1 are ~ 50%, while the levels of knockdown of SREBP2 are ~ 85% - are the authors sure that SREBP1 has no functional role? The authors must at least reach the same level of knockdown for both proteins before making their claims. More, over why do they use shRNAs against SREBP2 while they use siRNAs against SREBP1? Some

consistency in the experimental setup is also warranted as siRNAs/shRNAs have notorious off target effects;

4) In the Figure 4B, please show the effect of double silencing of BRD2 and SREBP2 on S2R mRNA and protein expression;

5) In Figure 7A, the only data presented is that BRD2 forms a complex with SREBP2; when this happens is also not clear as U18 treatment does not increase or decrease their binding capacity. Studying the activation state of SREBP2 (i.e. phosphorylation) in the assembly of the complex with BRD2 might be helpful;

6) In Figure 7A please include the control of SREBP1nT-FLAG construct and not FLAG-GFP;

7) In Figure 7A, controls testing whether SREBP2-nT does not co-IP with the other BRDs must be performed; again SBREBP1-nT must be included to reaffirm the specificity, as the authors claim, of their interaction;

8) In Figure 7B please indicate precisely the coordinates of the regions of BRD2 binding at S2R promoter in the ChIP experiment;

9) In Figure 7, there is no evidence in the paper of histone marks. Please show the modulation of different histone acetylation markers (i.e. H3K27Ac) at S2R promoter gene to corroborate the ChIP data;

10) In Figure 7B, please show the ChIP of SREBP2 at S2R promoter and show the co-occupancy with BRD2;

11) In Figure 7C the quality of the images presented is not satisfying. Please show a better image and quantify the amount of co-localization of the two markers in the nucleus;

12) In Figure S1A - a higher quality image is required, maybe even a single cell high resolution representative image to show lysosomal localization as the authors suggest. Alternatively, please add a marker (LAMP1) to show the accumulation of cholesterol in lysosomes;

13) In Figure S2B - BRD4 silencing is ~35% - 50%, this is a poor knockdown efficiency to interpret the function of BRD4. Why don't the authors use the BRD4 siRNA reported in Figure 3C for these experiments?

14) In Figure S3B why GAPDH shows 2 bands?

15) In Figure S3B first panel - FLAG-GFP has 2 bands whereas only 1 band is present in panels 2 and 3.

16) In general, some blots are marked with MW markers, while others are not. Please mark ALL blots with respective MWs for consistency throughout the manuscript.

We are very thankful to reviewers for the time and intellectual input they dedicated to our manuscript. We respect your decision based on their comments. While many of the reviewer#2's comments are very helpful, some reflect misperception of the real focus and scope of this manuscript, as we figured, likely due to the lack of clarity in our writing. That said, while some of their questions have to be experimentally addressed, we believe some can be explained. We feel that this manuscript may deserve a better opportunity, for the following main reasons.

1. Both reviewers are interested in and also recognize the novelty and importance of this work.
2. Reviewer#1 asserted "The results are clear and the conclusion is supported by the data". We agree that although our conclusion is solid, some experiments suggested by reviewers will enhance it.
3. Some of the major concerns raised by reviewer#2 have actually been already addressed in the literature, e.g. the functional effect of U18666A, or some are off the scope of this study. Again, we apologize that we did not make these points clear enough.
4. We figured that reviewer#2's some concerns are influenced by the appearance of Western blots. In fact, SREBP proteins inherently show multiple bands because their functions are associated with protein cleavage that is essential for their activation. For each Western blot figure, we typically performed at least 4 separate (different times) repeat experiments and the final data are quantified and statistically analyzed using appropriate methods which reviewers did not question. We thus believe that the robustness of the data is not a problem.

Please see below our responses to reviewer comments point by point (after >>). We keenly request that you may please communicate with the two reviewers on our responses, and hopefully you can reconsider our manuscript so that we can start performing the necessary experiments included in the responses.

Reviewer #1 (Comments to the Authors (Required)):

The authors investigated the transcriptional regulation of sigma-2 receptor (S2R) by the BET family protein BRD2. They showed that S2R transcript and protein levels increase upon cholesterol deprivation by an inhibitor of Niemann-Pick disease, type C1 (NPC1), a lysosomal cholesterol transporter and starvation. This S2R upregulation is abolished by the BET inhibitor JQ1 and BRD2 knockdown, while knockdown of other BET proteins, BRD3 and BRD4, has no effect on S2R levels. BRD2 forms a complex with SREBP2 and binds to the promoter region of S2R gene, which provide a mechanistic insight into the S2R upregulation upon treatment of the NPC1 inhibitor U18666A and starvation. Consistently, SREBP2 knockdown attenuates S2R expression in cells treated with U18666A and under starvation.

The results are clear and the conclusion is supported by the data. Their findings that BRD2 responses cholesterol deprivation are novel and potentially advance our understanding of the regulation of cholesterol homeostasis.

>>We appreciate the reviewer's assertion.

I have a couple of comments to strengthen the manuscript.

1. It is good to show that BRD2 does not bind to the further distal region of the S2R gene promoter.

>> This is a great suggestion. We will do the experiment.

2. They should conduct ChIP assay to show SREBP2 is recruited to the promoter region of S2R gene.

>> Yes, we will do the experiment.

3. They should examine the interaction between BRD2 and the full-length or endogenous SREBP2.

>> We will try. We did not perform this experiment considering that the full-length SREBP2 is an inactive form in the cytosol and hence not expected to interact with BRD2 in the nucleus.

4. U188666A should be U18666A.

>> We will correct. Thanks.

Reviewer #2 (Comments to the Authors (Required)):

"BRD2 regulation of sigma-2 receptor expression upon cytosolic cholesterol deprivation".

SUMMARY

The authors discovered an epigenetic circuit not previously reported, which is triggered by fluctuations in cytoplasm cholesterol levels. They report the new role for BET chromatin remodelers in the regulation of the expression of mRNA and protein of S2R, involved in the control of cholesterol homeostasis. BRD2 binds the transcription factor SREBP2 and both localize at the S2R gene promoter promoting its transcription.

The emerging role of S2R as a regulator of intracellular cholesterol levels in physiology and pathology, make important the understanding of the transcriptional/epigenetic programs controlling its expression, which represent a potential pharmacological target for the treatment of cholesterol-related dysfunctions and tumors.

>> We thank the reviewer for recognizing the importance of this work.

However, although the manuscript flow can be improved, I see major and minor points that need to be addressed for this manuscript to be published.

MAJOR POINTS

1) The authors show the effect of cholesterol deprivation from the cytoplasm (might be better to state from ER) on the induction of S2R expression. There are two gaps here that remain unresolved: i) what is the sensor that detects the accumulation of cholesterol in lysosomes, or the deprivation from ER (SCAP for the ER?) and activates the epigenetic/ transcriptional circuit resulting in S2R expression;

>> We apologize for the lack of clarity of writing that may have led to the reviewer's misunderstanding of the focus of this study. The real focus of this manuscript is on the BRD2 regulation of S2R's transcription. Moreover, this gap has been partially resolved in the literature (will elaborate below); on the other hand, addressing this gap takes a whole new study which is out of the scope of the current study.

ii) what is the biological meaning of S2R upregulation in ER cholesterol deprivation. They do not show any effect of the induction of the BRD2 / SREBP2 / S2R axis on uptake, total levels and on the distribution of intracellular cholesterol or LDLR.

It is clear that this is a paper on the transcriptional regulation of S2R,

>> We are thankful that the reviewer points out the real focus of the manuscript.

but adding further details about the two mentioned tasks, or at least discuss it better would be an added value for the paper.

>> Thanks for the suggestion. We will do experiments to check the effect of S2R silencing on cellular cholesterol levels or distribution.

2) The whole study was carried out by silencing the components of the disclosed circuit and no over-expression or rescue experiment was conducted.

>> This is a good point. We will perform gain-of-function over-expression experiments.

3) Serum starvation is not an efficient way for total cholesterol deprivation. Playing with complete medium, a cholesterol-free medium, in combination with HMG CoA reductase and U18 inhibitors would be useful to get selective ER deprivation, accumulation in lysosomes or total cholesterol abrogation.

>> We apologize that we did not explain clearly in the manuscript. We used serum starvation not intending to generate an efficient way for cholesterol deprivation because U18666A is efficient enough. Instead, we used this condition to differentiate the responses of SREBP1 (very responsive) and SREBP2 (not responsive, so is S2R).

4) All experiments in blocking cholesterol output from lysosomes were conducted with the use of U18. It would be advisable to silencing NPC's transporters or use NPC1 mutant cells to see if phenocopy the drug effect.

>> We apologize for lack of clarity. In fact, U18666A as a blocker of cholesterol output from lysosomes by targeting NPC1 has been established by Drs Brown and Goldstein (eLife 2015 PMID26646182).

5) U18 treatment of SCAP silenced cells might help in understanding the circuit better.

>> Since the current study aims to address how BRD2 regulates S2R transcription, we will cite literature that documents the signaling circuit involving SCAP responding to cholesterol deprivation (e.g. PMID28841344).

6) The quality of Co-IPs experiments is not satisfying and in the supplementary information. These represent the most important data of the paper.

>> Please see our responses to the reviewer's specific, substantiated critiques on this issue below in "minor points", from 5 to 11.

7) The authors didn't dig in to the intrinsic complexity of the epigenetic circuit. As in the case of Figure 3F in which the role of BRD4 in the regulation of S2R expression is not well understood. An important experiment that would be the overexpression (OE) of BRD2, 3 and 4 in absence of cholesterol deprivation. The BRD2-OE might indicate its sufficiency for the expression of S2R. Another key experiment would be OE of BRD4 in combination with BRD2 silencing. Furthermore, the epistatic experiment of the double silencing of BRD2 and 4 underlines: i) a role for BRD2 in the expression of S2R; ii) a role of BRD4 in the expression of S2R through induction of BRD2 expression; iii) a role for BRD4, independent of BRD2, in the repression of

S2R. In a paper studying the players involved in the regulation of S2R expression, the role of BRD4 in the repression of S2R cannot be neglected.

>> We agree with the reviewer on that OE gain-of-function experiments should be helpful and we will add these experiments.

8) A figure with a graphical abstract would support the readers. Authors should provide a model of their findings.

>> Great suggestion. Thanks.

9) In my opinion, this paper is not of satisfying quality, some experiments do not have proper controls

>> Please see our response to the reviewer's critique specific on the proper control issue below in "minor points".

and some blots and images are of very limited representative significance.

>> We will be able to replace these with blots that are more representative of the data in plots.

The data are very preliminary and some of the Western blots are not of satisfying quality.

>> The focus of this work is transcriptional regulation of S2R by BRD2. The gaps identified by the reviewer, such the target of U18, and the signaling circuit involving SCAP, have been addressed in the literature. We will make the writing more clear.

As for Western blots, please see our response to the reviewer's specific critique below in "minor points".

In terms of scientific impact, the authors claim to have found a transcription factor complex that regulates the expression of S2R when lysosomal cholesterol levels are perturbed. No washout or rescue experiments to show reversibility of the phenotype have been performed throughout the paper.

>> Thanks for the suggestion. Washout experiment will be performed.

I believe that to carry out all the aforementioned experiments it is not feasible in a standard time of a revision, and at present the paper is in a preliminary state for publication.

>> We feel that it is feasible in a reasonable time frame to accomplish only the necessary experiments to strengthen the conclusion on the focus of this manuscript, i.e. the BRD2 transcriptional regulation of S2R.

MINOR POINTS

1) In Figure 1D the blot is of not satisfying quality for S2R. In fact, going by that blot, under JQ1 treatment results in a complete loss of S2R protein levels. I find it fascinating that the authors claim this is non-significant in their bar graphs. How did they quantify this blot?

>> Thanks for the suggestion, we can replace the blot.

2) In Figure 3 please try overexpression of the different BRDs and check the effect on S2R mRNA and protein expression;

>> Yes, we can do, as stated above.

3) In Figure 4, the levels of knock down of SREBP1 are ~ 50%, while the levels of knockdown of SREBP2 are ~ 85% - are the authors sure that SREBP1 has no functional role? The authors must at least reach the same level of knockdown for both proteins before making their claims. More, over why do they use shRNAs against SREBP2 while they use siRNAs against SREBP1? Some consistency in the experimental setup is also warranted as siRNAs/shRNAs have notorious off target effects;

>> It would be a concern if S2R changed to the same direction after silencing SREBP1 and SREBP2. However, whereas silencing SREBP2 reduced S2R, silencing SREBP1 increased S2R.

4) In the Figure 4B, please show the effect of double silencing of BRD2 and SREBP2 on S2R mRNA and protein expression;

>> Yes, we will perform this experiment.

5) In Figure 7A, the only data presented is that BRD2 forms a complex with SREBP2; when this happens is also not clear as U18 treatment does not increase or decrease their binding capacity. Studying the activation state of SREBP2 (i.e. phosphorylation) in the assembly of the complex with BRD2 might be helpful;

>> We observed that U18 treatment slightly but not dramatically increased BRD2 co-IP compared to no U18 (Figure S3). We interpret that it is likely BRD2 Co-IP was almost saturated (because of overexpression of SREBP2nT) and hence U18 was unable to increase the co-IP much more.

6) In Figure 7A please include the control of SREBP1nT-FLAG construct and not FLAG-GFP;

>> This is a good point. We actually thought it over earlier. However, although our data showed that SREBP1 did not regulate S2R transcription, this does not preclude the possibility of SREBP1 interacting with BRD2 in the regulation of other non-S2R target genes. Therefore, we feel that SREBP1nT-FLAG is not necessarily a negative control and hence not a better control than FLAG-GFP which is commonly used by others.

7) In Figure 7A, controls testing whether SREBP2-nT does not co-IP with the other BRDs must

be performed; again SBREBP1-nT must be included to reaffirm the specificity, as the authors claim, of their interaction;

>> For the reason similar to the above explained, the result that S2R transcription is not regulated by other BRDs does not preclude the possibility for BRD3/4 to interact with SREBP2 in regulating genes other than S2R.

8) In Figure 7B please indicate precisely the coordinates of the regions of BRD2 binding at S2R promoter in the CHIP experiment;

>> Yes. Thanks.

9) In Figure 7, there is no evidence in the paper of histone marks. Please show the modulation of different histone acetylation markers (i.e. H3K27Ac) at S2R promoter gene to corroborate the CHIP data;

>> Since it is off the focus of this study, we don't feel that the experiment on histone acetylation is very helpful to strengthen the main conclusion of this manuscript. Moreover, a sobering reality is that there are multiple acetylation sites on H3 alone, and BRD2 could bind H4 as well where there are multiple acetylation sites. Thus, figuring out the acetylation site(s) per se take a whole new study project.

10) In Figure 7B, please show the CHIP of SREBP2 at S2R promoter and show the co-occupancy with BRD2;

>> Agreed. We will do the experiments as we answered to reviewer#1 question 2.

11) In Figure 7C the quality of the images presented is not satisfying. Please show a better image and quantify the amount of co-localization of the two markers in the nucleus;

>> We can try, but this is pretty much the limit of confocal imaging for the nucleus. Moreover, since BRD2 has more broad functions, we don't expect a complete overlapping/co-localization of BRD2 and SREBP2.

12) In Figure S1A - a higher quality image is required, maybe even a single cell high resolution representative image to show lysosomal localization as the authors suggest. Alternatively, please add a marker (LAMP1) to show the accumulation of cholesterol in lysosomes;

>> U18-induced cholesterol accumulation in lysosomes has been well established and proven by LAMP1 staining in published reports (e.g. PMID29530923; PMID21074609). We will make this point more clear in the manuscript and cite these papers.

13) In Figure S2B - BRD4 silencing is ~35% - 50%, this is a poor knockdown efficiency to interpret the function of BRD4. Why don't the authors use the BRD4 siRNA reported in Figure 3C for these experiments?

>> We indeed used siRNA which showed robust knockdown of BRD4 (Figure 3C). Figure S2B is the same experiment except for using a secondary approach, shRNA.

14) In Figure S3B why GAPDH shows 2 bands?

>> Thanks. We can replace this blot.

15) In Figure S3B first panel - FLAG-GFP has 2 bands whereas only 1 band is present in panels 2 and 3.

>> Given long enough exposure, a second minor band would appear.

16) In general, some blots are marked with MW markers, while others are not. Please mark ALL blots with respective MWs for consistency throughout the manuscript.

>> We will fix. Thanks.

MS: LSA-2019-00540

Dr. Lian-Wang Guo
The Ohio State University
473 West 12th Ave
Columbus Columbus

Dear Dr. Guo,

Thank you for your recent correspondence regarding our decision on your manuscript "BRD2 regulation of sigma-2 receptor expression upon cytosolic cholesterol deprivation". As mentioned in our initial decision letter, we would be open for resubmission should you be able to address the reviewer concerns. Overall, we appreciate your outline and we came to the conclusion that we can encourage you to revise your work for resubmission to Life Science Alliance. However, some aspects need to get addressed in a slightly better way:

- the experiments of IP and ChIP must be performed with the appropriate controls to corroborate the specificity of SREBP2 and BRD2 in the transcriptional regulation of S2R after cholesterol deprivation by the ER.
- the silencing approach needs to be consistent (either siRNA or shRNA and with satisfactory silencing levels)
- the quality of blots and IF images need to be significantly improved
- silencing of NPC transporters and subsequent analyses is needed as an additional line of evidence (instead of only having the inhibitor based assays)
- to address reviewer #2's minor point 5 you should try to improve the SREBP2Nt transfection (SREBP2Nt barely visible in the input in fig S3) to increase the dynamic range of the interaction with the endogenous BRD2 and to study better if the addition of U18 improves the interaction

Please note that I also consulted with reviewer #2 to give you more definitive feedback on the outline you provided and that this reviewer only supports further consideration if the revision addresses the above points.

Yours sincerely,

Reviewer #1 (Comments to the Authors (Required)):

The authors investigated the transcriptional regulation of sigma-2 receptor (S2R) by the BET family protein BRD2. They showed that S2R transcript and protein levels increase upon cholesterol deprivation by an inhibitor of Niemann-Pick disease, type C1 (NPC1), a lysosomal cholesterol transporter and starvation. This S2R upregulation is abolished by the BET inhibitor JQ1 and BRD2 knockdown, while knockdown of other BET proteins, BRD3 and BRD4, has no effect on S2R levels. BRD2 forms a complex with SREBP2 and binds to the promoter region of S2R gene, which provide a mechanistic insight into the S2R upregulation upon treatment of the NPC1 inhibitor U18666A and starvation. Consistently, SREBP2 knockdown attenuates S2R expression in cells treated with U18666A and under starvation.

The results are clear and the conclusion is supported by the data. Their findings that BRD2 responses cholesterol deprivation are novel and potentially advance our understanding of the regulation of cholesterol homeostasis.

I have a couple of comments to strengthen the manuscript.

>>Thank you for your review and comments.

1. It is good to show that BRD2 does not bind to the further distal region of the S2R gene promoter.

>> Thank you. We appreciate the suggestion on more negative control data. We thus performed new experiments to add another negative control in addition to IgG, please see Figure 9C, and corresponding revision in Page 6, Paragraph 3:

“Moreover, we performed the same ChIP experiment but to detect a different site (~200 bp from TSS) in the S2R gene promoter that is low-ranked in SRE prediction. As seen in Figure 9C, the qPCR signal did not respond to U18 treatment or BRD2 expression, thus providing another negative control for the methodology in addition to IgG.”

2. They should conduct ChIP assay to show SREBP2 is recruited to the promoter region of S2R gene.

>> Accomplished. Please see a new ChIP-qPCR experiment, which indicated SREBP2 recruitment to the promoter of S2R gene. Please find Figure S4 and corresponding revision in Page 5, Paragraph 2:

“Indeed, ChIP-qPCR assay consistently showed SREBP2Nterm occupancy at the S2R gene promoter (Figure S4)”.

3. They should examine the interaction between BRD2 and the full-length or endogenous SREBP2.

>> This suggestion was respectively well taken. Since the full-length SREBP2 localizes in the ER and BRD2 is in the nucleus, their interaction has to occur with the SREBP2 N-terminal half which is able to translocate into the nucleus. Unfortunately, thus far there hasn't been a good commercial antibody available for the endogenous SREBP2 N-terminal half. We tried 4 antibodies from different sources but none worked. This is the sole reason for that we have to work with tagged SREBP2 N-terminal half for its interaction with endogenous BRD2.

4. U188666A should be U18666A.

>> Corrected. Thank you.

Reviewer #2 (Comments to the Authors (Required)):

"BRD2 regulation of sigma-2 receptor expression upon cytosolic cholesterol deprivation".

SUMMARY

The authors discovered an epigenetic circuit not previously reported, which is triggered by fluctuations in cytoplasm cholesterol levels. They report the new role for BET chromatin remodelers in the regulation of the expression of mRNA and protein of S2R, involved in the control of cholesterol homeostasis. BRD2 binds the transcription factor SREBP2 and both localize at the S2R gene promoter promoting its transcription.

The emerging role of S2R as a regulator of intracellular cholesterol levels in physiology and pathology, make important the understanding of the transcriptional/epigenetic programs controlling its expression, which represent a potential pharmacological target for the treatment of cholesterol-related dysfunctions and tumors.

>> Thank you for your comments.

However, although the manuscript flow can be improved, I see major and minor points that need to be addressed for this manuscript to be published.

MAJOR POINTS

1) The authors show the effect of cholesterol deprivation from the cytoplasm (might be better to state from ER) on the induction of S2R expression. There are two gaps here that remain unresolved: i) what is the sensor that detects the accumulation of cholesterol in lysosomes, or the deprivation from ER (SCAP for the ER?) and activates the epigenetic/ transcriptional circuit resulting in S2R expression; ii) what is the biological meaning of S2R upregulation in ER cholesterol deprivation. They do not show any effect of the induction of the BRD2 / SREBP2 / S2R axis on uptake, total levels and on the distribution of intracellular cholesterol or LDLR.

It is clear that this is a paper on the transcriptional regulation of S2R, but adding further details about the two mentioned tasks, or at least discuss it better would be an added value for the paper.

>> The reviewer is correct. The real focus of this manuscript is the BRD2 transcriptional regulation of S2R. Thank you. Please see our detailed responses below and corresponding revisions in the manuscript. To help readers to understand the context, we also added more information with additional literature cited pertaining to the two mentioned tasks. For example, Page 5, Paragraph 2:

“It is established that SREBP2 in complex with the SREBP-cleaving activation protein (SCAP) can sense a decrease of ER cholesterol and translocate to Golgi, where the SREBP2 N-terminal half (abbreviated as SREBP2Nterm) is cleaved off and then able to enter the nucleus to bind sterol regulatory elements (SREs) of genomic DNA, thereby regulating gene expression.”

And Page 7, Paragraph 3:

“Recent studies indicated that S2R co-localizes with LDLR, which via internalization carries esterified cholesterol into the cell³; S2R knockdown or knockout impairs cholesterol (and LDLR) uptake.”

And Page 9, Paragraph 1:

“This novel result is consistent with the well-documented SREBP2 functional mechanism; i.e. SREBP2 in an inactive state resides in the ER membrane, but upon drop of cytosolic/ER cholesterol levels, SREBP2 is transported to Golgi and cleaved into two half molecules³. Whereas the C-terminal half stays in the cytosol, the N-terminal half enters the nucleus acting as a TF for the expression of genes involved in cholesterol metabolism and transport”.

2) The whole study was carried out by silencing the components of the disclosed circuit and no over-expression or rescue experiment was conducted.

>> We performed new over-expression experiments. Please see Figures 3 and 4.

3) Serum starvation is not an efficient way for total cholesterol deprivation. Playing with complete medium, a cholesterol-free medium, in combination with HMG CoA reductase and U18 inhibitors would be useful to get selective ER deprivation, accumulation in lysosomes or total cholesterol abrogation.

>> The reviewer’s point is well taken. U18666A’s role was recently established by the Brown and Goldstein labs as targeting NPC1 to block cholesterol from exiting lysosomes (eLife 2015 PMID26646182). Moreover, our data showed a clear effect of U18666A on cholesterol entrapment in lysosomes. We therefore used U18666A as the primary method to induce ER cholesterol deprivation and in parallel used serum starvation as a secondary method. The imaging data illustrating U18666’s effect is now presented in Figure 1, I and J. Please also see more information added, in Page 4, Paragraph 1:

“We then used U18666A (abbreviated as U18 throughout) as a tool to generate an experimental condition for cholesterol deprivation, as it is an established NPC1 inhibitor that keeps cholesterol trapped inside the lysosome thereby producing an intracellular environment with cholesterol reduced in the endoplasmic reticulum (ER)”.

And Page 4, Paragraph 1:

“Confirming the function of U18 in reducing cholesterol in the cytosol and hence endoplasmic reticulum (ER)^{3, 18}, filipin staining of cholesterol disappeared from most of the intracellular space (or the ER network) after U18 treatment but instead accumulated in perinuclear structures (Figure 1, I and J), which are typically known to be lysosomes.”

4) All experiments in blocking cholesterol output from lysosomes were conducted with the use of U18. It would be advisable to silencing NPC1 transporters or use NPC1 mutant cells to see if phenocopy the drug effect.

>> Thank you. We actually tested NPC1 silencing, which did not produce a robust effect on inducing S2R upregulation as U18 did. Please see Figure S2. A possible reason is that NPC1 silencing cannot completely deplete NPC1 protein, whereas its inhibitor U18 could fully block its function in lysosomal cholesterol export.

5) U18 treatment of SCAP silenced cells might help in understanding the circuit better.

>> Thanks for the reminder. To help readers to understand the circuit, we have now cited the literature that establishes the cholesterol-sensing circuit involving SCAP. Please see the revision in Page 5, Paragraph 2:

“It is established that SREBP2 in complex with the SREBP-cleaving activation protein (SCAP) can sense a decrease of ER cholesterol and translocate to Golgi, where the SREBP2 N-terminal half (abbreviated as SREBP2Nterm) is cleaved off and then able to enter the nucleus to bind sterol regulatory elements (SREs) of genomic DNA, thereby regulating gene expression”.

6) The quality of Co-IPs experiments is not satisfying and in the supplementary information. These represent the most important data of the paper.

>> This has been improved. We now have representative immunoblots in Figure 9A and Figure S7, and also the original blots included in the supplemental file.

7) The authors didn't dig in to the intrinsic complexity of the epigenetic circuit. As in the case of Figure 3F in which the role of BRD4 in the regulation of S2R expression is not well understood. An important experiment that would be the overexpression (OE) of BRD2, 3 and 4 in absence of cholesterol deprivation. The BRD2-OE might indicate its sufficiency for the expression of S2R. Another key experiment would be OE of BRD4 in combination with BRD2 silencing. Furthermore, the epistatic experiment of the double silencing of BRD2 and 4 underlines: i) a role for BRD2 in the expression of S2R; ii) a role of BRD4 in the expression of S2R through induction of BRD2 expression; iii) a role for BRD4, independent of BRD2, in the repression of S2R. In a paper studying the players involved in the regulation of S2R expression, the role of BRD4 in the repression of S2R cannot be neglected.

>> Accomplished. We did new experiments and the data are presented in Figures 3, 4, and S7. The results further confirmed a specific role of BRD2 in regulating S2R expression.

Please see a whole new paragraph added at the bottom of Page 4:

“To further delineate a BRD2-specific role in governing S2R expression, we performed gain-of-function experiments....”

8) A figure with a graphical abstract would support the readers. Authors should provide a model of their findings.

>> Great suggestion. Thanks. Please see the new Figure 12.

9) In my opinion, this paper is not of satisfying quality, some experiments do not have proper controls and some blots and images are of very limited representative significance. The data are very preliminary and some of the Western blots are not of satisfying quality.

>> We have performed new experiments and fixed these problems to address the reviewer’s each concern. Please see our point-by-point responses above and below.

In terms of scientific impact, the authors claim to have found a transcription factor complex that regulates the expression of S2R when lysosomal cholesterol levels are perturbed. No washout or rescue experiments to show reversibility of the phenotype have been performed throughout the paper.

>> We have added new data of overexpression of BRD2, 3, and 4, as stated above, and now the reversibility of phenotype is demonstrated by our loss and gain of function experiments. In addition, further confirming the specificity of JQ1, its enantiomer did not show (whereas JQ1 did) a function in reversing the S2R-upregulating effect of U18. Please see Figure S3 and revision in Page 4, Paragraph 1:

“The JQ1’s enantiomer, which is a chemically identical yet functionally inert stereoisomer hence an ideal negative control for JQ1, did not significantly alter S2R expression (Figure S3), confirming JQ1 effect’s specificity for BETs.”

I believe that to carry out all the aforementioned experiments it is not feasible in a standard time of a revision, and at present the paper is in a preliminary state for publication.

>> Even though with the tremendous challenges imposed by both the COVID-19 pandemic and lab relocation, we strived to complete the suggested many experiments to address each of the reviewers’ comments.

MINOR POINTS

1) In Figure 1D the blot is of not satisfying quality for S2R. In fact, going by that blot, under JQ1

treatment results in a complete loss of S2R protein levels. I find it fascinating that the authors claim this is non-significant in their bar graphs. How did they quantify this blot?

>> Fixed. Please see the new Figure 1D.

2) In Figure 3 please try overexpression of the different BRDs and check the effect on S2R mRNA and protein expression;

>> Accomplished. Please see the new Figures 3 and 4 and a whole new paragraph in Page 4 (bottom), and our responses to the reviewer comments (Major question #7) above.

3) In Figure 4, the levels of knock down of SREBP1 are ~ 50%, while the levels of knockdown of SREBP2 are ~ 85% - are the authors sure that SREBP1 has no functional role? The authors must at least reach the same level of knockdown for both proteins before making their claims.

>> Efficiency of siRNAs varies depending on the commercial availability of products. In the case of SREBP siRNAs, it would be a concern if S2R levels had changed to the same direction after silencing SREBP1 and SREBP2. However, whereas silencing SREBP2 reduced S2R, silencing SREBP1 increased S2R, thus a change to the opposite direction occurred.

More, over why do they use shRNAs against SREBP2 while they use siRNAs against SREBP1? Some consistency in the experimental setup is also warranted as siRNAs/shRNAs have notorious off target effects;

>> Fixed. We have performed new experiments and changed all to siRNAs for consistency. Please see Figures 5, 6, and 8.

4) In the Figure 4B, please show the effect of double silencing of BRD2 and SREBP2 on S2R mRNA and protein expression;

>> Accomplished. Thanks. Please see the new Figure 6, and the revision in Page 5, Paragraph 3:

“Now that the above results showed that both BRD2 and SREBP2 were positive regulators of S2R expression, we next used combined siRNAs for BRD2 and SREBP2 double silencing. The data (Figure 6) demonstrated that BRD2/SREBP2 double silencing nearly completely blocked S2R protein production. Therefore, up to this point, our results had provided compelling evidence for BRD2 being a positive regulator of S2R mRNA and protein expression.”

5) In Figure 7A, the only data presented is that BRD2 forms a complex with SREBP2; when this happens is also not clear as U18 treatment does not increase or decrease their binding capacity. Studying the activation state of SREBP2 (i.e. phosphorylation) in the assembly of the complex with BRD2 might be helpful;

>> This is a good point. Thanks. In fact, the N-terminal half molecule of SREBP2 used in this study is its active form, as the C-terminal half inhibits its transcription factor activity in the full-length SREBP2 protein. We did observe that U18 treatment increased BRD2/SREBP2Nterm co-IP though not significantly by statistics (Figure 9B). We interpret that the lack of a dramatic effect of U18 on enhancing the co-IP may stem from SREBP2Nterm overexpression which could have saturated its binding with BRD2 even without U18. Please see the revision in Page 6, Paragraph 2:

“the BRD2/SREBP2Nterm co-IP was further enhanced by U18 treatment, although the change did not reach a statistical significance, reflecting a likelihood that BRD2 was nearly saturated by the binding of overexpressed SREBP2Nterm.”

6) In Figure 7A please include the control of SREBP1nT-FLAG construct and not FLAG-GFP;

>> Thanks for the suggestion. We actually thought it over earlier. However, although our data showed that SREBP1 is not the primary regulator of S2R transcription, this does not preclude the possibility of SREBP1 interacting with BRD2 to regulate other target genes. In fact, we found that SREBP1 co-IPed with SREBP2Nterm/BRD2 (Figure S7). Therefore, we reason that SREBP1nT-FLAG is not qualified to serve as a negative control.

7) In Figure 7A, controls testing whether SREBP2-nT does not co-IP with the other BRDs must be performed; again SBREBP1-nT must be included to reaffirm the specificity, as the authors claim, of their interaction;

>> Accomplished. We did new experiments, which indicated that SREBP2-nT (i.e. SREBP2Nterm) co-IPed BRD2 (as also shown in Figure 9A), but not BRD4. Please see Figure S7 and corresponding revision in Page 6, the middle of Paragraph 2:

“BRD2 (Figure 9, A and B) but not BRD4 (Figure S7) co-IPed with SREBP2Nterm;”

8) In Figure 7B please indicate precisely the coordinates of the regions of BRD2 binding at S2R promoter in the ChIP experiment;

>> Done. It is now specifically indicated in Figure 9C and legends. Please also see the revision in Page 6, Paragraph 3:

“Finally, we used a BRD2 antibody for IP and performed ChIP-qPCR to detect S2R promoter regions that contain predicted SREs or consensus SREBP-binding motifs. We found that while U18 treatment increased qPCR signal of a S2R promoter region (~1000 bp from TSS) by > 2 fold, increasing BRD2 further augmented the signal, either in the absence or presence of U18 (Figure 9C).”

9) In Figure 7, there is no evidence in the paper of histone marks. Please show the modulation of different histone acetylation markers (i.e. H3K27Ac) at S2R promoter gene to corroborate the ChIP data;

>> Accomplished. Our new experiments indicated occupancy of H3K27Ac at the S2R gene promoter. Please see the new Figure 10, and revision in Page 6, Paragraph 3:

“It is known that BRD2 through its bromodomains binds the epigenomic mark H3K27Ac to facilitate transcriptional activation at select gene loci. We thus further corroborated the observed BRD2 occupancy of S2R promoter via ChIP-qPCR using a H3K27Ac-specific antibody. Consistently, the data indicated H3K27Ac-enriched S2R promoter DNA.”

10) In Figure 7B, please show the ChIP of SREBP2 at S2R promoter and show the co-occupancy with BRD2;

>> Accomplished. Our new ChIP-qPCR experiments showed SREBP2 occupancy at S2R promoter (the site of ~1000 bp from TSS), the same region where BRD2 occupies. Please see Figure S4, and revision in Page 5, Paragraph 2:

“Indeed, ChIP-qPCR assay consistently showed SREBP2Nterm occupancy at the S2R gene promoter (Figure S4).”

11) In Figure 7C the quality of the images presented is not satisfying. Please show a better image and quantify the amount of co-localization of the two markers in the nucleus;

>> Accomplished. We performed new experiments for confocal imaging. Please see the new Figure 11 and corresponding revision at the bottom of Page 6:

“In addition, we performed fluorescence imaging to illustrate protein subcellular distribution. As shown by Figure 11, GFP or mCherry alone (not in fusion with BRD2 or SREBP2Nterm) evenly dispersed throughout the whole cell, indicative of a non-specific distribution pattern. However, both BRD2 and SREBP2Nterm, whether tagged with GFP or mCherry, were confined in the nucleus with high level of overlap, verifying proper nuclear localization of these ectopically expressed proteins.”

12) In Figure S1A - a higher quality image is required, maybe even a single cell high resolution representative image to show lysosomal localization as the authors suggest. Alternatively, please add a marker (LAMP1) to show the accumulation of cholesterol in lysosomes;

>> Accomplished. We are now presenting high-resolution images of cholesterol lysosomal localization. Please see Figure 1, I and J, and revision in Page 4, Paragraph 1:

“Confirming the function of U18 in reducing cholesterol in the ER, filipin staining of cholesterol disappeared from most of the intracellular space (or the ER network) after U18 treatment but instead accumulated in perinuclear structures (Figure 1, I and J), which are typically known to be lysosomes.”

13) In Figure S2B - BRD4 silencing is ~35% - 50%, this is a poor knockdown efficiency to interpret the function of BRD4. Why don't the authors use the BRD4 siRNA reported in Figure 3C for these experiments?

>> Accomplished. We did new experiments and changed shRNAs to siRNAs throughout which showed robust silencing efficiency. Please see Figures 1 and 2.

14) In Figure S3B why GAPDH shows 2 bands?

>> Fixed.

15) In Figure S3B first panel - FLAG-GFP has 2 bands whereas only 1 band is present in panels 2 and 3.

>> Fixed.

16) In general, some blots are marked with MW markers, while others are not. Please mark ALL blots with respective MWs for consistency throughout the manuscript.

>> Fixed. Thank you.

October 26, 2020

RE: Life Science Alliance Manuscript #LSA-2019-00540RR-A

Dr. Lian-Wang Guo
University of Virginia
409 Lane Rd, MR4 (building), Room 2146
Charlottesville, VA 22908

Dear Dr. Guo,

Thank you for submitting your revised manuscript entitled "BRD2 regulation of sigma-2 receptor upon cholesterol deprivation". We would be happy to publish your paper in Life Science Alliance pending final revisions necessary to meet our formatting guidelines.

Along with the points listed below, please also attend to the following:

- please consult our Manuscript Preparation Guidelines <https://www.life-science-alliance.org/manuscript-prep> and put your manuscript sections in the correct order
- please add ORCID ID for corresponding author
- please provide your manuscript text as editable doc format
- please add Author Contributions to main manuscript text
- please upload both your main and supplementary figures as single files
- please add a separate section for your main and supplementary figure legends in your main manuscript text rather than having the figure legends below each figure
- please add a callout for Figure 4A in your main manuscript text
- please improvise the layout of Figure 7 to make it more seamless (A, B, C..) and edit the figure legend text to make it clearer as to what each panels within the figure represent
- please improvise the legends for Figure S1 and S8 Panels to clarify what the panels within these figures represent
- please provide source data for gels Figures 1B, 7H, and 9A.

A. FINAL FILES:

B. MANUSCRIPT ORGANIZATION AND FORMATTING:

Sincerely,

Shachi Bhatt, Ph.D.
Executive Editor

Reviewer #1 (Comments to the Authors (Required)):

The authors properly addressed the comments raised by this reviewer. Therefore, I recommend its publication.

Reviewer #2 (Comments to the Authors (Required)):

"BRD2 regulation of sigma-2 receptor expression upon cytosolic cholesterol deprivation".

SUMMARY

The authors have discovered an epigenetic circuit that has not been previously reported, which is triggered by fluctuations in cytoplasm cholesterol levels. They report the new role for BET chromatin remodelers in the regulation of mRNA and protein expression of S2R, which controls cholesterol homeostasis. Briefly, BRD2 binds SREBP2 transcription factor, the interaction is tuned by cholesterol levels in ER, and both localize at the S2R gene promoter promoting its transcription.

The emerging role of S2R as a regulator of intracellular cholesterol levels in physiology and pathology is well documented. Thus marking the importance of studying the transcriptional/epigenetic programs controlling its expression. Further, representing a potential pharmacological target for the treatment of cholesterol-related dysfunctions and tumors.

MAJOR POINT

The authors have presented several revised experiments to strengthen their findings. Specifically, they now show publication quality blots and Co-IP experiments. They have performed over-expression and rescue experiments which highlight the specificity of BRD2 as a regulator of S2R transcription. This has been corroborated by ChIP experiment of SREBP2 and H3K27Ac at the S2R promoter. The authors could have made an extra effort to improve the silencing of NPC1 transporter as trigger of S2R expression. Further, the inefficient silencing of SREBP1, and the complex between BRD2-SREBP2 and SREBP1 as demonstrated by Co-IP still leaves some doubts that it could also be involved in the expression of S2R.

However, this observation does not change the importance of the novel findings and the possible pharmacological impact. The authors now have enough data to support their statements and, I suggest publishing the manuscript in your journal.

November 2, 2020

RE: Life Science Alliance Manuscript #LSA-2019-00540RRR

Dr. Lian-Wang Guo
University of Virginia
409 Lane Rd, MR4 (building), Room 2146
Charlottesville, VA 22908

Dear Dr. Guo,

Thank you for submitting your revised manuscript entitled "BRD2 regulation of sigma-2 receptor upon cholesterol deprivation" to Life Science Alliance (LSA). There are still a couple issues remaining that need to be addressed before we can move forward with your manuscript.

- Could you please share with us the original unprocessed blots for Figures 1B, 7H, and 9A?
- Could you please improve the figure legends for S8 and call out the S8 panels in the corresponding legend?
- Could you please also make sure that the manuscript sections are in appropriate order, in accordance to the LSA formatting guidelines - www.life-science-alliance.org/manuscript-prep

A. FINAL FILES:

B. MANUSCRIPT ORGANIZATION AND FORMATTING:

Sincerely,

Shachi Bhatt, Ph.D.
Executive Editor
Life Science Alliance
<https://www.lsjournal.org/>
Tweet @SciBhatt @LSAJournal

November 11, 2020

RE: Life Science Alliance Manuscript #LSA-2019-00540RRRR

Dr. Lian-Wang Guo
University of Virginia
409 Lane Rd, MR4 (building), Room 2146
Charlottesville, VA 22908

Dear Dr. Guo,

Thank you for submitting your Research Article entitled "BRD2 regulation of sigma-2 receptor upon cholesterol deprivation". It is a pleasure to let you know that your manuscript is now accepted for publication in Life Science Alliance. Congratulations on this interesting work.

DISTRIBUTION OF MATERIALS:

Again, congratulations on a very nice paper. I hope you found the review process to be constructive and are pleased with how the manuscript was handled editorially. We look forward to future exciting submissions from your lab.

Sincerely,

Shachi Bhatt, Ph.D.

Executive Editor

Life Science Alliance

<https://www.lsjournal.org/>
